# Controlling synthetic membraneless organelles by a red-light-dependent singlet oxygen-generating protein

Manjia Li [1], Byung Min Park [1], Xin Dai[2,3], Yingjie Xu[1,4], Jinqing Huang [2] & Fei Sun [1,4,5,6 ✉]

Membraneless organelles (MLOs) formed via protein phase separation have great implications for both physiological and pathological processes. However, the inability to precisely control the bioactivities of MLOs has hindered our understanding of their roles in biology, not to mention their translational applications. Here, by combining intrinsically disordered domains such as RGG and mussel-foot proteins, we create an *in cellulo* protein phase separation system, of which various biological activities can be introduced via metal-mediated protein immobilization and further controlled by the water-soluble chlorophyll protein (WSCP)—a remarkably stable, red-light-responsive singlet oxygen generator. The WSCP-laden protein condensates undergo a liquid-to-solid phase transition on light exposure, due to oxidative crosslinking, providing a means to control catalysis within synthetic MLOs. Moreover, these photoresponsive condensates, which retain the light-induced phase-transition behavior in living cells, exhibit marked membrane localization, reminiscent of the semi-membrane-bound compartments like postsynaptic densities in nervous systems. Together, this engineered system provides an approach toward controllable synthetic MLOs and, alongside its light-induced phase transition, may well serve to emulate and explore the aging process at the subcellular or even molecular level.

[1] Department of Chemical and Biological Engineering, The Hong Kong University of Science and Technology, Clear Water Bay, Kowloon, Hong Kong, China. [2] Department of Chemistry, The Hong Kong University of Science and Technology, Clear Water Bay, Hong Kong, China. [3] Laboratory for Synthetic Chemistry and Chemical Biology, Health@InnoHK, Hong Kong Science Park, Hong Kong, China. [4] Greater Bay Biomedical InnoCenter, Shenzhen Bay Laboratory, Shenzhen 518036, China. [5] Biomedical Research Institute, Shenzhen Peking University–The Hong Kong University of Science and Technology Medical Center, Shenzhen 518036, China. [6] HKUST Shenzhen Research Institute, Shenzhen 518057, China. ✉email: kefsun@ust.hk

Eukaryotic cells use various organelles confined by lipid membranes to ensure spatiotemporal control over intracellular activities. In recent years, membraneless organelles (MLOs) have also gained traction with biologists. These non-canonical subcellular microcompartments — often the products of liquid-liquid phase separation (LLPS) of biomacromolecules—can selectively concentrate proteins and/or nucleic acids in specific foci while allowing free exchange of molecules within and without[1,2]. Accumulating evidence suggests that these condensates can fulfill a variety of physiological functions. Typical examples include P bodies in the cytoplasm for translation regulation, Cajal bodies in the nucleus for genome organization, and postsynaptic density (PSD) in the synapse for scaffolding and signaling[3–5].

The formation of phase-separated condensates is also highly relevant to disease pathologies. Pathological aggregation, a type of aberrant phase separation, of intrinsically disordered proteins (IDPs) such as FUS, TDP-43, and Tau is believed to be critical for neurodegenerative disorders and tumorigenesis[6–8]. The protein condensates may transition from dynamic liquid droplets to static solid particles with the passage of time and/or in response to environmental stress, a process reminiscent of material aging[9,10]. However, despite the tremendous efforts made to understand them under physiological conditions, the exact roles in these condensates in physiology and pathology, as well as the mechanisms by which they are regulated, remain elusive.

A variety of optogenetic tools have been developed, enabling delicate control within complex biological systems. Among the first are rhodopsin proteins, a collection of light-gated ion channels or pumps that have been engineered for controlling electrical and biochemical events in a light-dependent manner[11]. Another group of widely used optogenetic tools is derived from plant or bacterial photoreceptor proteins, such as cryptochrome 2 (Cry2), light-oxygen-voltage-sensing domain (LOV), and Dronpa, which undergo conformational changes on light exposure, often accompanied with homo- or hetero-oligomerization as well. These molecular tools have been adopted for modulating biological signaling, regulating gene expression, controlling intracellular phase separation, and even creating smart biomaterials[12,13]. However, an obvious drawback of these optogenetic tools lies in their dependence on short-wave-length light (e.g., blue light), which could be troublesome insofar as cytocompatibility and deep-tissue penetration are concerned. Efforts have recently been shifted to develop red- or far-red-light-dependent optogenetic tools, such as those based on plant or bacterial phytochromes, PhyB/PIF and BphS, for biological regulation and biomaterial design[14–17]. However, the bulky size of PhyB (~99 kDa) and the complexity of BphS, the latter involving a cascade of c-di-GMP-mediated signaling events, have prevented their wider use inside living cells. It is therefore desirable to further enlarge the arsenal of optogenetic tools, especially the long-wavelength-light-dependent ones, by tapping into some different photochemistry.

Water-soluble chlorophyll proteins (WSCPs) from plants in the *Brassicaceae* family have been known as a group of cytoplasmic photosensitizers capable of generating singlet oxygen on light exposure, while their true physiological roles remain mysterious[18]. A typical apoWSCP is monomeric and tetramerizes upon binding to chlorophyll (Fig. 1). More importantly, unlike most chlorophyll proteins that are membrane-bound, the WSCPs are cytoplasmic and highly soluble in water. These photosensitizers can absorb red light and convert ground-state oxygen ($^3O_2$) into singlet oxygen ($^1O_2$)[18.] As reactive oxygen species (ROSs) are involved in various biological processes, ranging from nerve degeneration and aging to innate immunity, we envisioned that this red-light-induced singlet oxygen generation by WSCPs

might provide an alternative approach for controlling biochemical events at the cellular or subcellular level, whether natural or synthetic.

In this study, we created an optically controllable MLO system via the LLPS of a recombinant protein comprising a tyrosine-rich mussel foot protein 3 (Mfp-3) domain flanked by two highly positively charged arginine-glycine-glycine repeat (RGG) domains. The resulting protein condensates can be readily decorated by bioactive molecules such as WSCP via hydrophobic effect and fluorescent proteins or caspase-3—a key player in apoptosis—via His6-tag/metal coordination. The ensuing red-light-induced generation of singlet oxygen by WSCP facilitated oxidative crosslinking within the protein condensates, triggered their liquid-to-solid phase transition, and ultimately shut down the biochemical activities within. Moreover, these protein condensates, with marked membrane localization, retained the photo-induced phase transition behavior inside living cells. This study illustrates a powerful strategy for controlling biochemical events, and even emulating aging processes, at the subcellular or even molecular level.

## Results

**Design of protein construct capable of LLPS**. We designed the gene encoding a multi-domain recombinant protein, RGG-Mfp-3-RGG (RMR) (Fig. 2a). This protein could serve to construct membraneless organelles for two chief reasons. Firstly, the intrinsically disordered RGG domain, which is prevalent and evolutionarily conserved among a variety of protein isoforms, is noted for its concentration- and temperature-dependent phase separation behavior[19–21]. Secondly, mussel foot proteins (Mfps) from *Mytilus edulis* have been widely known for their central roles in underwater adhesion, a phenomenon which is often attributed to their unique amino acid compositions, i.e., the high proportion of DOPA—a noncanonical amino acid derived from oxidation of Tyr—and positively charged Lys and Arg, as well as their robust phase separation behavior on a variety of substrates;[22–25] the Mfp-3 used in this study, when produced heterologously by *Escherichia coli*, is abundant in Tyr, which is prone to oxidation and thus provides a opportunity for mechanical tuning via oxidative crosslinking (Supplementary Fig. 1). It is therefore conceivable that the combination of the redox-responsive Mfp-3 motif and the RGG motif, both characterized by robust phase separation, could serve to create a smart, dynamically tunable MLO system.

We produced the RMR protein using an *E. coli* expression system and purified it with Ni-NTA chromatography. The purified protein exhibited a reversible phase separation behavior in a temperature-dependent manner in vitro, in line with the previous finding that the RGG motif has an upper-critical solution temperature (UCST);[19] protein condensates emerged as the temperature dropped from 52 to 23 °C and disappeared rapidly as the temperature returned to 52 °C (Fig. 3b). Moreover, these condensates readily coalesced into larger ones at 23 °C, suggesting dynamic, liquid-like properties (Fig. 3d and Supplementary Movie 1). The absorbance at 600 nm, a measurement of turbidity, increased as the RMR concentration did, indicative of concentration-driven phase separation (Fig. 2b, c).

**Modulation of RMR condensates by WSCP**. Rapid coalescing of the RMR condensates suggests a high surface tension between water and these droplets, likely resulting from the hydrophobicity of the unstructured RMR (Fig. 3b). Decorating these condensates with hydrophilic molecules might help reduce the surface tension and stabilize them in aqueous solutions. The WSCP, a tetrameric chlorophyll-binding globular protein originated from *Lepidium*

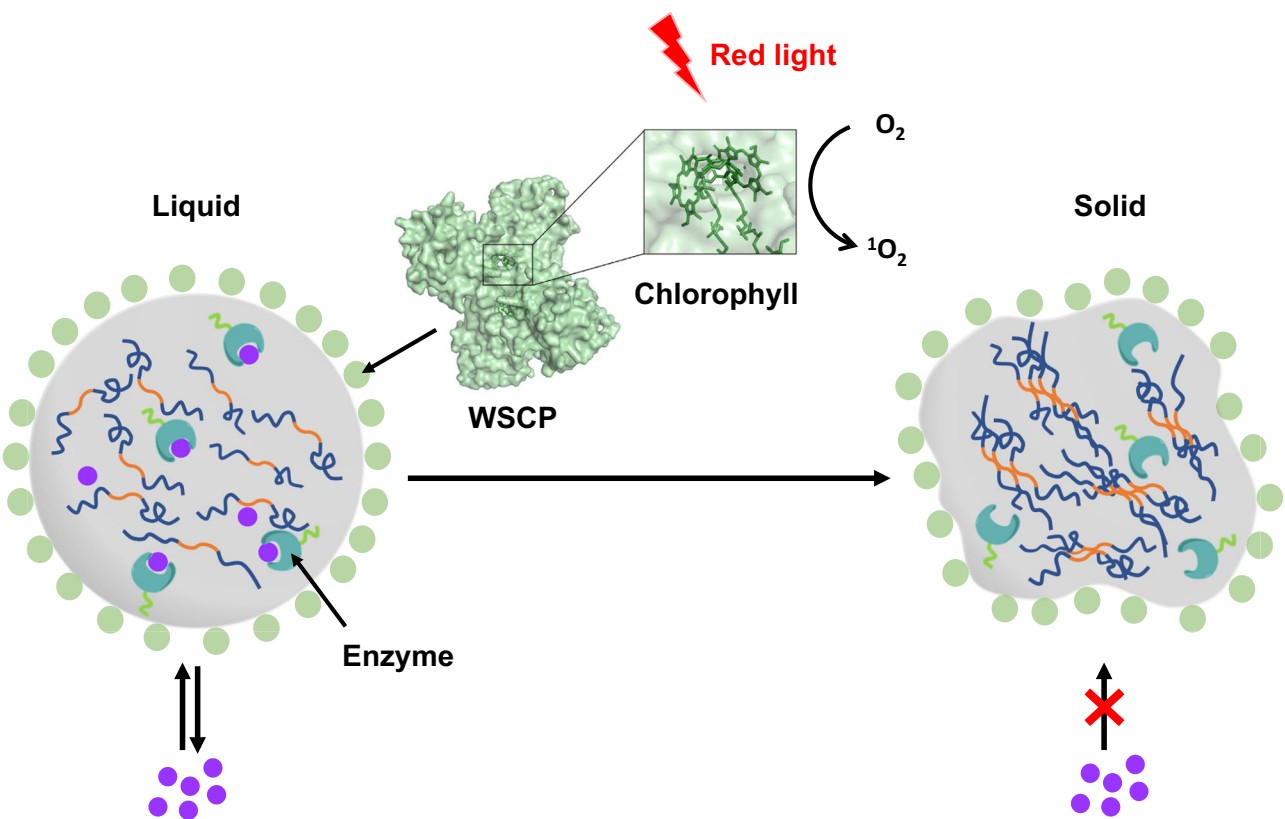

**Fig. 1 Schematic illustration of a red-light-controlled protein condensate enabled by water-soluble chlorophyll-binding protein (WSCP).** The protein condensate formed via LLPS of RGG-Mfp-3-RGG (RMR) is decorated with WSCP (PDB: 2DRE), a tetrameric protein that generates singlet oxygen ($^1O_2$) under red light irradiation. Upon photooxidation, the protein condensate undergoes a liquid-to-solid phase transition, which restricts the diffusion of substrates into the enzyme-laden protein condensate and turns off the catalysis within.

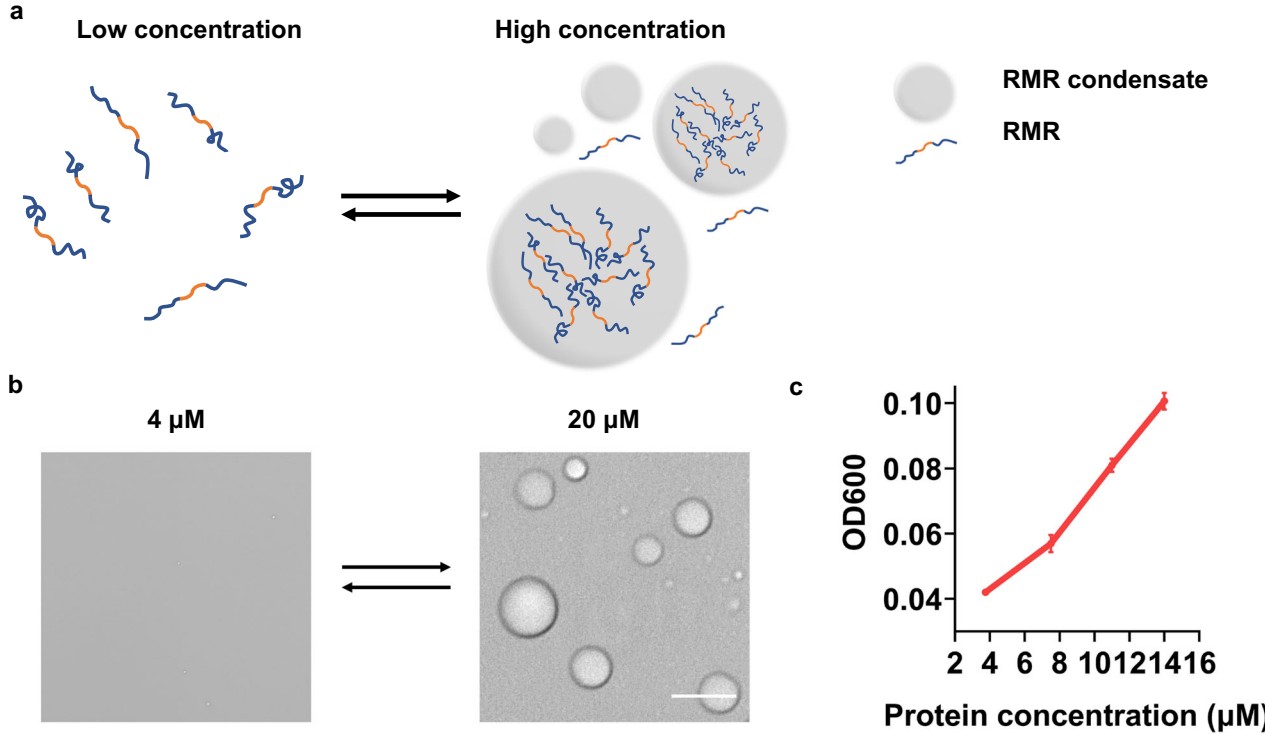

**Fig. 2 Concentration-dependent liquid-liquid phase separation of RMR. a** Schematic showing the phase separation of RMR. **b** Concentration-dependent phase separation of RMR at room temperature (23 °C). Scale bar: 5 µm. Images representative of $n = 3$. **c** Optical densities at 600 nm at different RMR concentrations. The threshold RMR concentration for phase separation at 23 °C is 4 µM. Data are presented as mean ± SD ($n = 3$).

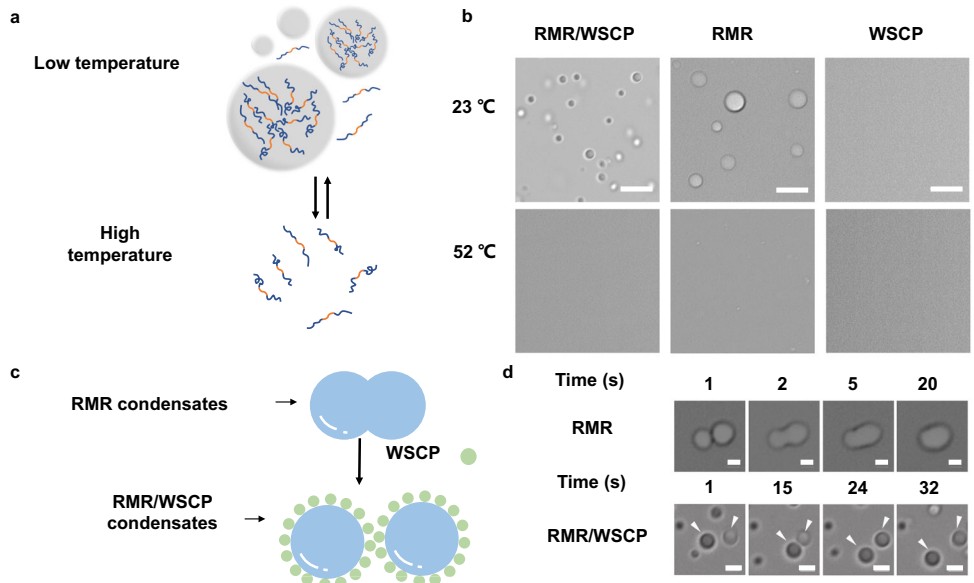

**Fig. 3 Temperature-dependent liquid-liquid phase separation of RMR in the presence and absence of WSCP. a** Schematic showing temperature-dependent phase separation of RMR. **b** Representative images showing temperature-dependent phase separation of RMR in the presence and absence of WSCP. Scale bars: 5 μm. Images representative of $n = 3$. **c** Schematic showing stabilization of RMR condensates by WSCP. Two RMR condensates are coalescing into a bigger one in the absence of WSCP, while remaining separate upon decoration with WSCP. Confocal imaging analysis revealed the presence of WSCP both on the surfaces and inside of the RMR condensates (Supplementary Fig. 4). The cartoons of WSCP inside the condensates are omitted for clarity. **d** Representative images showing RMR condensates in the absence and presence of WSCP at 23 °C with the passage of time. Scale bars: 1 μm.

*virginicum*, is highly soluble in aqueous solutions and possesses a rather hydrophilic surface[18,26,27]. We successfully produced recombinant WSCP in its chlorophyll-bound form using *E. coli* expression and a single-step reconstitution, where the *E. coli* cells were mixed with spinach juice—the source of chlorophyll—and homogenized prior to the protein purification. The reconstituted WSCP was purified from the cell lysates first by Ni-NTA chromatography, and then by size-exclusion chromatography (SEC) to ensure its purity. The resulting protein solution was green and exhibited strong absorbance at 400–500 nm and 650–700 nm, characteristic of a chlorophyll-binding protein (Supplementary Figs. 2 and 3).

To examine the influence of WSCP on the RMR condensates, we added WSCP (5 μM) into the phase-separated RMR system, followed by gentle pipetting to achieve thorough mixing. The resulting condensates remained dynamic and thermally sensitive; they disappeared at an elevated temperature (52 °C), similar to those in the absence of WSCP (Fig. 3b). On the other hand, these condensates turned out to be more uniform in size and more stable than those in the absence of WSCP; they did not coalesce with each other even in proximity, which is highly indicative of reduced surface tension, presumably because of the decoration by hydrophilic WSCP (Fig. 3d and Supplementary Movie 2). The intrinsic fluorescence of WSCP, arising from the chlorophyll cofactor, enabled us to confirm its interaction with the RMR condensates via confocal microscopy[28]. We observed an obvious enrichment of WSCP by the RMR condensates, likely due to hydrophobic effect (Supplementary Fig. 4); WSCP was present not only on the surfaces but also inside of the condensates. Although quantitative analysis on the exact distribution of WSCP within a given condensate remains challenging, because of the small size of the condensate (<5 μm in diameter) and the limited resolution of the confocal microscope, it is conceivable that the WSCPs on the surfaces of these condensates are likely to be the major contributors to their reduced surface tension and increased stability.

As WSCP has been known to be able to generate singlet oxygen under the irradiation of red light (620–660 nm)[29–34], we envisioned that the introduction of WSCP would enable not only physical but also chemical modulation of the protein condensates; light-induced generation of singlet oxygen by WSCP would oxidize and crosslink the unstructured, Tyr- and Trp-rich RMR polypeptides[35,36]. Indeed, after brief red-light irradiation (5 mW/cm²) for ~1 min, some RMR/WSCP condensates gradually transformed in shape, from uniform, smooth-surface microspheres to irregularly shaped, shrunken particles, highly indicative of the occurrence of a liquid-to-solid phase transition (Fig. 4b and Supplementary Movie 3). Within 30 min, almost all the condensates deswelled and transitioned into solid-like particles (Fig. 4c and Supplementary Fig. 5). The observed deswelling reflected the contribution from interchain crosslinking among protein polymers, which was further confirmed by sodium dodecyl sulfate–polyacrylamide gel electrophoresis (SDS-PAGE) analysis of the RMR condensates (Supplementary Fig. 6)[37]. This liquid-to-solid phase transition was accompanied with a 3-fold increase in fluorescence intensity ($\lambda_{ex} = 280$nm, $\lambda_{em} = 400$nm), pointing to the possible formation of fluorescent moieties due to oxidative crosslinking of the aromatic residues, while the nature of these fluorophores has yet to be determined (Supplementary Fig. 7)[38].

To confirm that it is the WSCP-bound chlorophyll, rather than free chlorophyll dissociating from the protein, that contributed to the observed oxidative crosslinking, we performed SEC analysis on the reconstituted WSCP. It turned out that the protein was predominantly tetrameric, in agreement with the formation of the chlorophyll/WSCP complex (Supplementary Fig. 8)[18]. In addition, this complex has previously been shown to be remarkably stable toward dissociation and protein denaturation even at 100 °C and extreme pH values (pH 0–14)[18]. Together, these results rule out the possibility that free chlorophyll could dissociate from the WSCP complex under the ambient conditions (i.e., room temperature, pH 7-8) used in this study, not to

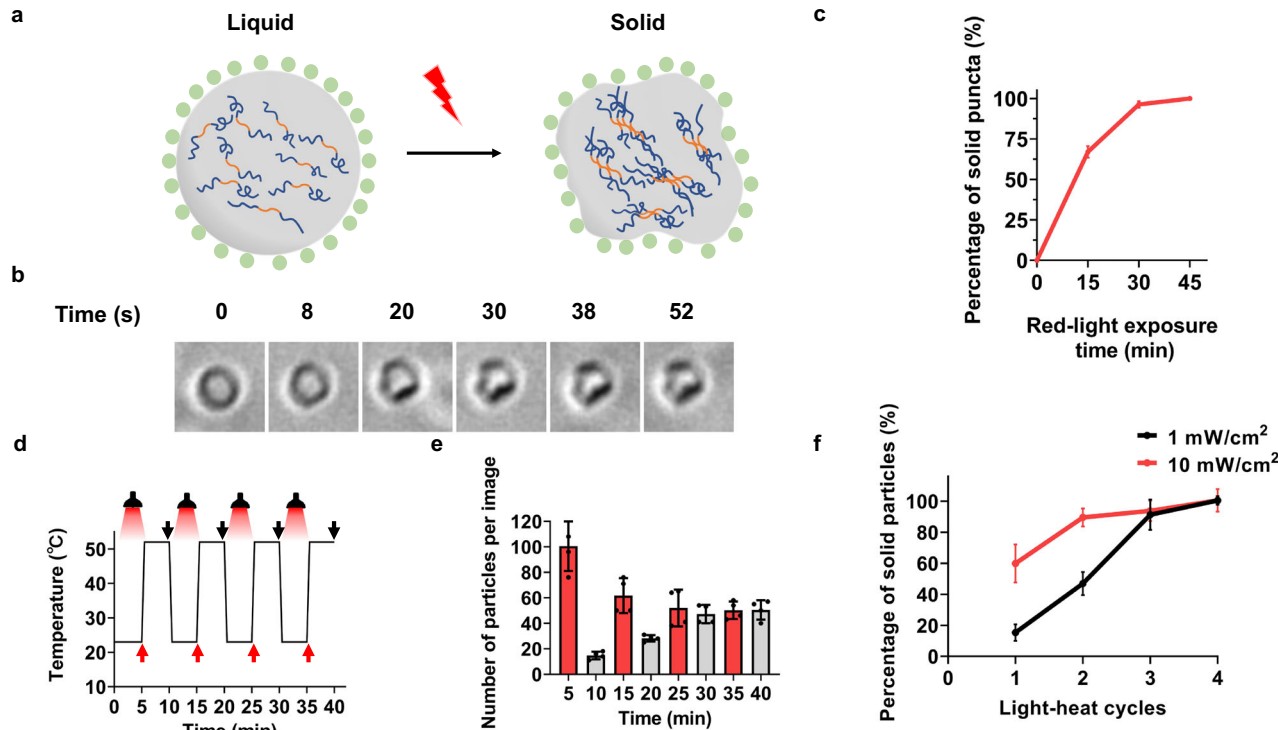

**Fig. 4 Red-light-dependent liquid-to-solid phase transition of RMR/WSCP condensates. a** Schematic illustration of red-light-induced liquid-to-solid phase transition process. **b** Appearance of an RMR/WSCP condensate at 23 °C under irradiation with the passage of time. Images representative of $n > 10$. **c** Percentages of solid particles throughout red-light irradiation with the passage of time. Solid particles were differentiated from the spherical liquid condensates based on their irregular shapes (Supplementary Fig. 3). Data are presented as mean ± SD ($n = 3$). **d** Schematic showing light-heat cycling experiments that comprise iterative red-light irradiation at 23 °C (5 min, 1 mW/cm²) and ensuing incubation at 52 °C in the dark (5 min). Images were taken at time points as indicated by red and black arrows. **e** Number of solid particles observed per image. Data are presented as mean ± SD ($n = 4$). **f** Percentage of heat-resistant solid particles observed after each light-heat cycle. Data are presented as mean ± SD ($n = 4$).

mention the generation of singlet oxygen by any unbound cofactor.

Contrary to the liquid condensates before irradiation, which dissipated rapidly upon heating, the solid particles after irradiation became resistant to heat, which remained intact at the elevated temperature (52 °C), as expected for covalently crosslinked solid materials (Supplementary Fig. 9). The Mfp-3 domain proved to be essential for the light-induced phase transition, as the irradiated condensates comprising RGG-CarHC-RGG (RCR)/WSCP, with Mfp-3 replaced with the folded globular CarHC[39], remained thermally sensitive (Supplementary Fig. 9). Intriguingly, throughout the light-heat cycling experiments with varied light irradiation, the induced liquid-to-solid phase transition was characterized by stochasticity, despite relative uniformity and simplicity of the RMR/WSCP condensates in size, geometry, and physicochemical composition (Fig. 4d). When weak light irradiation (1 mW/cm²) was used, only a minor portion (~15%) of the condensates transitioned from liquid to solid and remained after the first light-heat cycle (Fig. 4e, f and Supplementary Fig. 10). In the ensuing cycles, cooling to 23 °C led to the reemergence of liquid protein condensates, though decreasing in number, which subsequently transitioned into solid particles under continuous light irradiation in a stochastic manner. After four cycles, all the condensates became heat-resistant, suggesting a complete liquid-to-solid transition (Fig. 4e, f and Supplementary Fig. 10). Light intensity affects the efficiency of the liquid-to-solid phase transition, presumably via the rate of generating singlet oxygen. Increasing the irradiation power from 1 to 10 mW/cm² facilitated the transition from liquid condensates into solid particles; under the enhanced irradiation, it took merely

two light-heat cycles to complete the phase transition (>90% solid particles) (Fig. 4f and Supplementary Fig. 11). The light-induced, singlet oxygen-mediated liquid-to-solid phase transition of the RMR/WSCP condensates is reminiscent of the pathological protein aggregation in vivo—the epitome of aging—in that the latter is often preceded by a physiological LLPS and accompanied with accumulating reactive oxygen species[40]. Some common features, such as stochasticity, heterogeneity, and gradualism, are also shared between material aging and biological aging, as both require continuous or repetitive cues such as oxidative stress, over a timespan largely decided by the intensity of the stress.

**Spatial modulation of RMR/WSCP condensates via optical tweezers.** Previous studies have established optical tweezers as a versatile tool for controlling biomacromolecules and LLPS with high spatial precision[41,42]. This technology enabled us to selectively trap two protein condensates in the presence of 5 μM WSCP, designated as Trap 1 and Trap 2, respectively. Irradiating the designated site of the chosen condensate (Trap 1) with an additional triggering laser beam subsequently triggered the liquid-to-solid phase transition in a controllable manner (Fig. 5c and Supplementary Movie 4). The trapped condensate (Trap 1) started to shrink—a sign of solidification—first at its irradiated site within 7 s and then across the entire condensate, resulting in an irregularly shaped solid particle within 15 s, while the other condensate (Trap 2) without the additional laser irradiation remained intact throughout the experiment. WSCP proved to be crucial for this precisely controlled phase transition, as no solidification was observed in the absence of WSCP even under

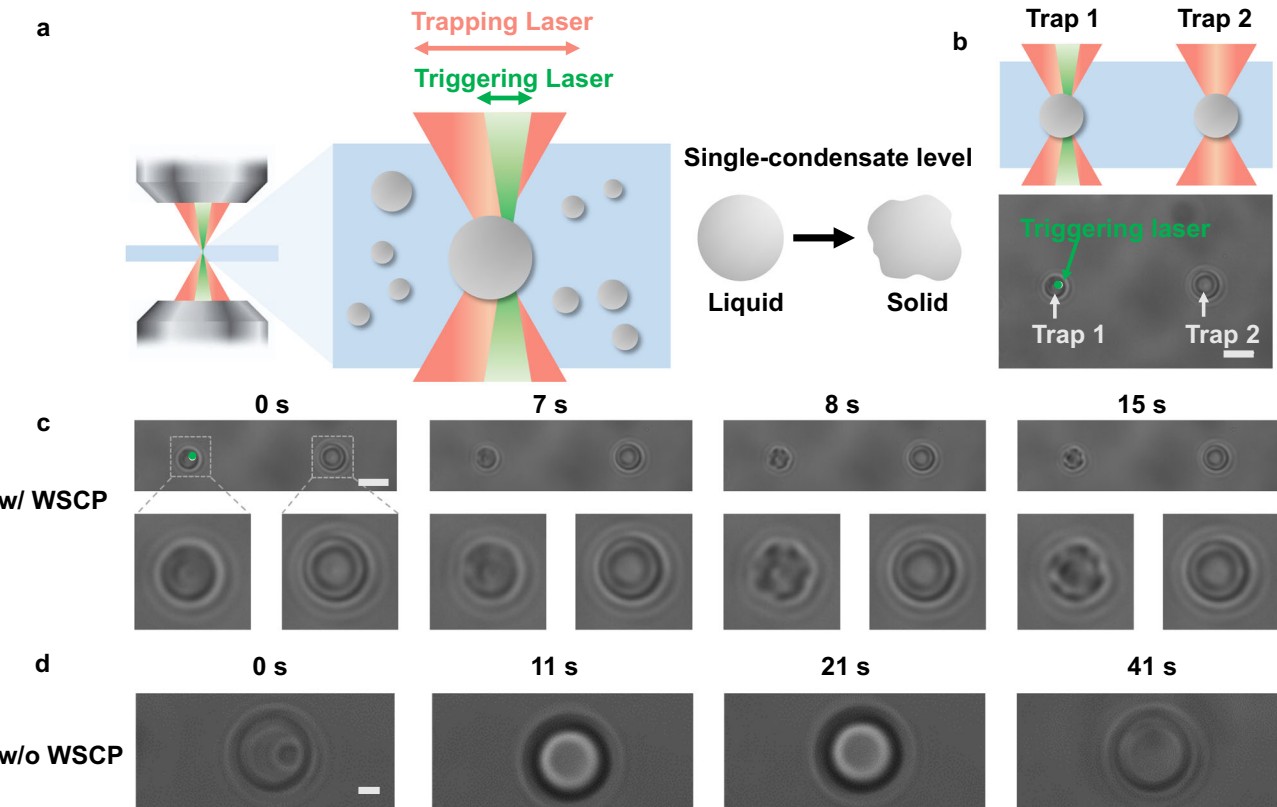

**Fig. 5 Modulation of RMR/WSCP condensates with high spatial precision by optical tweezers. a** Schematic illustration of the use of a modified optical tweezer system consisting of trapping and triggering laser beams to modulate the protein condensates at the single-condensate level. Trapping laser: 1064 nm; 80 mW. Triggering laser: 532 nm; 2.5 mW. **b** Schematic illustration of the trapping of two protein condensates (Trap 1 and Trap 2) by optical tweezers, followed by selective irradiation of the chosen condensate (Trap 1; left) via the triggering laser beam. A live camera image of two trapped condensates is shown. Scale bar: 5 µm. **c** Liquid-to-solid phase transition of the chosen RMR/WSCP condensate (*left*) under the irradiation of the triggering laser beam within 15 s. The concentration of WSCP is 5 µM. Scale bar: 5 µm. Images representative of $n = 9$. **d** The RMR condensate in the absence of WSCP is inert to prolonged irradiation. Scale bar: 1 µm. Images representative of $n = 2$.

prolonged irradiation (41 s) (Fig. 5d). Decreased concentration of WSCP (1 µM) led to decelerated phase transition (Supplementary Fig. 12 and Supplementary Movie 5). Together, these results demonstrated the possibility of optically modulating these protein condensates at the single-condensate level, with high spatio-temporal precision.

**Metal-mediated immobilization of protein cargos.** MLOs, formed via LLPS, often serve to compartmentalize biosynthetic pathways comprising one or several enzymes, spontaneously or in response to certain cues, for improved efficiency or selectivity. For example, the phase condensation of the complexes of cyclic GMP-AMP synthase and DNA, promoted by transition metal ions such as $Zn^{2+}$, $Mn^{2+}$ or $Co^{2+}$, boosts the production of the secondary messenger cGAMP and activates innate immune signaling[43].

Prompted by this metal-induced phase condensation, we envisioned the possible use of metal/polyhistidine-tag (or His6-tag) coordination as a general strategy to compartmentalize functional proteins into MLOs (Fig. 6a). The putative metal-binding motif, His6-tag, has been widely used in recombinant proteins for purification purposes. The coordination interactions between His6-tag and transition metal ions such as $Ni^{2+}$, $Zn^{2+}$, $Cu^{2+}$ or $Co^{2+}$ have also proven to be effective in assembling recombinant proteins into macroscopic hydrogel materials and mesoscopic or microscopic liquid condensates[44,45]. To examine the feasibility of recruiting functional proteins into the RMR condensates via the metal/His6-tag coordination, we mixed the

cargo molecules, GFP, with the RMR solution in the absence or presence of $Ni^{2+}$. It turned out that a pronounced amount of GFP molecules were enriched within the RMR condensates in the presence of $Ni^{2+}$ (50 µM), as evidenced by the fluorescence emitted, which is substantially stronger than (3.4 times) that of the surrounding solution (Fig. 6b, c). This drew a contrast with the systems free of $Ni^{2+}$ and treated with excess of the chelator, ethylenediaminetetraacetic acid (EDTA) (2 mM), where the GFP molecules were largely excluded from the condensates, resulting in a fluorescence intensity less than half (~40%) that of the surrounding solution. Moreover, other transition metal ions such as $Zn^{2+}$ also facilitated the recruitment of His6-tagged GFP into the RMR condensates (Supplementary Fig. 13). Among many possible interactions, the metal/ligand coordination is likely to be the chief one responsible for the interaction between the protein cargos and the condensates, given the latter's sensitivity to EDTA and dependence on metal ions like $Ni^{2+}$ and $Zn^{2+}$. In light of the prevalence of the His6-tag motif, alongside a good enrichment efficiency (~8.5 fold), the metal/His6-tag coordination indeed provided us with a generalizable and efficient way to compartmentalize functional recombinant proteins.

**Photo-controlled catalysis within RMR condensates.** In nature, subcellular organelles are beneficial to biosynthesis not only in efficiency; they also constitute a fundamental mechanism by which cellular organisms achieve controllability and specificity within complex systems[46]. To recapitulate the sophistication of the naturally occurring bioreactors, it is desirable to have the

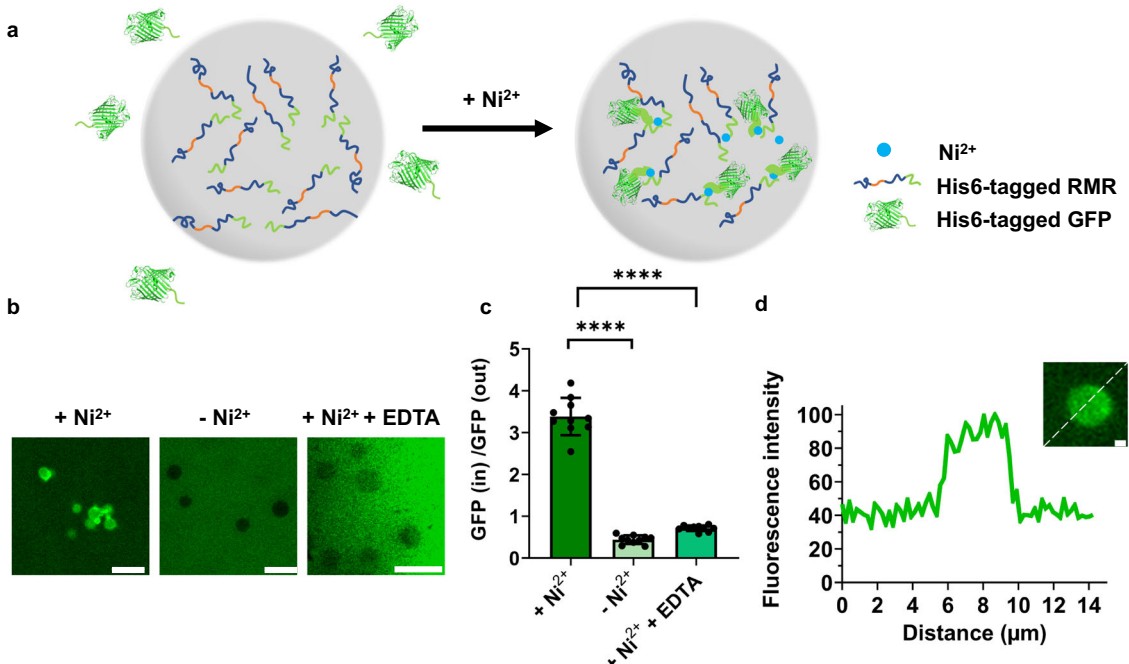

**Fig. 6 Decoration of RMR condensates with His6-tagged GFP via metal coordination. a** Schematic illustration of $Ni^{2+}$-induced recruitment of His6-tagged GFP into RMR condensates. **b** Fluorescence micrographs showing GFP-deprived and GFP-laden RMR condensates in the absence and presence of $Ni^{2+}$ (50 μM), as well as those in the presence of $Ni^{2+}$ (50 μM) and excess of EDTA (2 mM). Scale bars: 10 μm. Images representative of $n = 3$. **c** Relative distribution of GFP inside and outside RMR condensates. Enrichment efficiency was calculated using the equation, $Enrichment = GFP(in)/GFP(out)$. Data are presented as mean ± SD ($n = 10$); two-side $t$-test, $p$-value: **** $< 0.0001$ [$p = 2.75 \times 10^{-8}$ ($+ Ni^{2+}$ vs - $Ni^{2+}$); $p = 3.19 \times 10^{-5}$ ($+ Ni^{2+}$ vs. $+ Ni^{2+} + EDTA$)]. **d** Representative micrograph of a GFP-laden condensate and its normalized fluorescence intensity profile. Scale bar: 1 μm.

ability to design micro- or nano-structures with controllable reactivities.

The WSCP-dependent, light-induced liquid-to-solid phase transition system might provide us with the opportunity to design such controllable mini-reactors, in which chemical reactions can be modulated via altering the phase state of the reactors. To examine the feasibility of controlling catalysis via inducible phase transition, we chose as the model system caspase-3, a protease that hydrolyses its cognate peptide substrate, DEVD, and plays a central role in apoptosis[47,48]. Its enzymatic activity can be readily assessed with the fluorogenic substrate, Z-DEVD-AFC, which generates a fluorescent compound, 7-amino-4-trifluoromethylcoumarin (AFC), upon hydrolysis (Fig. 7b)[47,49].

His6-tagged caspase-3 was immobilized onto the RMR/WSCP condensates in the presence of $Ni^{2+}$, followed by irradiation under different light conditions. We subsequently assessed these condensates using the standard fluorescence-based caspase-3 activity assay (Supplementary Fig. 14). It turned out that exposure to red light (5 mW/cm²) for 2 h abolished the proteolytic activity of these WSCP-laden condensates, while exerting little influence over the condensates free of WSCP (Fig. 7c). For these WSCP-laden condensates, the inactivation of caspase-3 was proportional to light intensity and duration; lowered illumination intensity (1 mW/cm²) or shortened exposure time (≤1.5 h) led to moderate inhibition over the enzymatic reaction, pointing to the possibility of fine-tuning these protein-based micro-reactors with light (Fig. 7d).

We conjectured that the WSCP-dependent, photo-induced liquid-to-solid phase transition, which rendered the protein condensates impermeable and the caspase-3 within inaccessible to the DEVD substrates, might constitute the underlying mechanism for the decrease in proteolytic activity after light irradiation. To verify this and rule out the possibility that singlet oxygen may simply impair the caspase-3 activity by chemically

damaging the enzyme, we examined the influence of light irradiation on free caspase-3 in the presence of WSCP. After 30-min irradiation (5 mW/cm²), the activities of free caspase-3 were barely affected, suggesting the negligible influence of photo-oxidation on the free enzyme (Supplementary Fig. 15). Together, these results demonstrated the use of the light-induced phase transition to modulate substrate accessibility as a feasible strategy for controlling the chemical reactivity of synthetic organelles.

**RMR condensates in living cells**. To assess the phase separation behavior of RMR in living cells, we introduced into HEK293 and Hela cells the gene encoding RMR-mCherry. The protein indeed phase-separated in both cell lines (Fig. 8a, b and Supplementary Fig. 16). The Mfp-3 domain was crucial for the observed phase separation, as replacing it with GFP significantly diminished the percentage of cells that possessed puncta, from (73.4 ± 10.0)% to (2.9 ± 5.1)% (Fig. 8a, b).

RMR condensed not only in the cytoplasm but also, to a larger extent, along the cell membrane (Fig. 8c). A collection of $z$ slices obtained using confocal microscopy, alongside the reconstituted 3D image, revealed the enrichment and uneven distribution of RMR-mCherry along the cell membrane (Fig. 8c and Supplementary Movie 6). One possible explanation is that the coagulation of two oppositely charged polymers—the positively charged Mfp-3 domains and the negatively charged membrane lipids—facilitates the clustering and phase separation of RMR on cell membranes. It is corroborated by the fact that the RGG-GFP-RGG (RGR) protein exhibited significantly diminished abilities to cluster on the cell membranes, confirming the importance of the positively charged Mfp-3 domain for the observed protein/membrane interactions[50,51]. In addition, according to the fluorescence recovery after photobleaching (FRAP) assay, the puncta comprising RMR-mCherry in the cytoplasm and on the

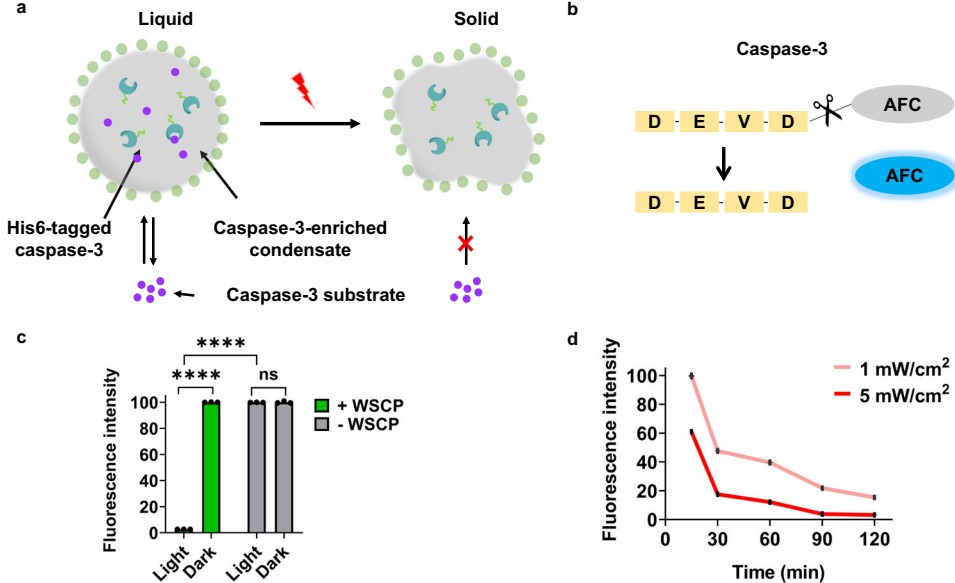

**Fig. 7 Photo-controlled catalysis within RMR condensates. a** Schematic illustration of the mechanism for controlling catalysis within the caspase-3-laden condensate via red-light-induced phase transition. Substrates readily diffuse into a liquid condensate, but not into a solid particle. **b** Schematic illustration of the fluorogenic caspase-3 substrate, Z-DEVD-AFC. AFC, 7-amino-4-trifluoromethylcoumarin, with $\lambda_{ex} = 405$ nm and $\lambda_{em} = 500$ nm. **c** Influence of light and WSCP on bound caspase-3. Light intensity: 5 mW/cm²; irradiation duration: 120 min. Data are presented as mean ± SD ($n = 3$); two-side $t$-test, $p$-value: **** < 0.0001 ($p = 3.41 \times 10^{-7}$ (Light + WSCP $vs$ Light ⁻WSCP)); $p = 7.92 \times 10^{-7}$ (Light + WSCP vs. Dark + WSCP). **d** Influence of irradiation duration and intensity on catalysis. Data are presented as mean ± SD ($n = 3$).

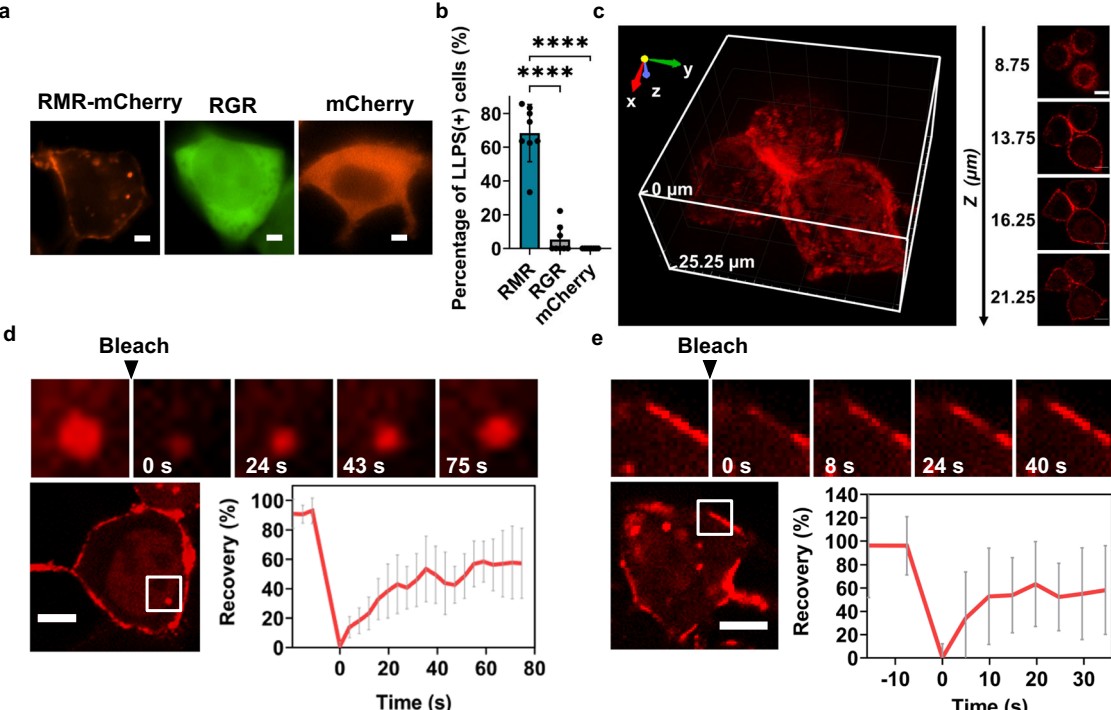

**Fig. 8 Phase separation in living cells. a** Fluorescent images of HEK293 cells transfected with the gene encoding RGG-Mfp-3-RGG-mCherry (RMR-mCherry), mCherry, or RGG-GFP-RGG (RGR). Scale bars: 5 μm. **b** Percentage of HEK293 cells that exhibit LLPS. Data are presented as mean ± SD ($n = 8$); two-side $t$-test, $p$-value: **** < 0.0001, $p = 9.30 \times 10^{-6}$ (RMR $vs$ RGR); $p = 8.89 \times 10^{-6}$ (RMR vs mCherry). **c** 3D rendering and representative $z$-slice images of HEK293 cells producing RMR-mCherry. Scale bars: 10 μm. FRAP assays of RMR condensates in the cytoplasm (**d**) and in the outer membrane of HEK293 cells (**e**). The plots show the normalized fluorescence recovery after photobleaching. Scale bars: 10 μm. Data are presented as mean ± SD ($n = 8$ in **d** and $n = 16$ in **e**).

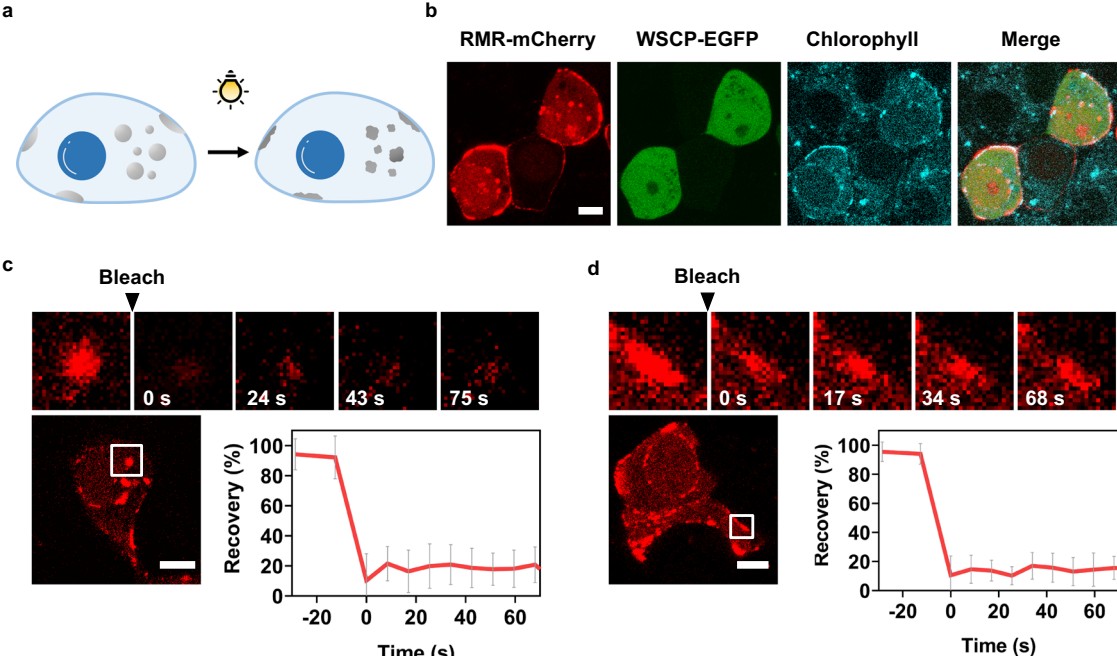

**Fig. 9 Light-induced liquid-to-solid phase transition in HEK293 cells. a** Schematic illustration of light-induced phase transition in cells. **b** Fluorescent images of the cells harbouring chlorophyll ($\lambda_{ex}$, 405 nm; $\lambda_{em}$, 640 nm) while expressing RMR-mCherry and EGFP-WSCP. Scale bar: 10 μm. Images representative of $n = 5$. FRAP assays of RMR condensates in the cytoplasm (**c**) and in the outer membrane of HEK293 cells (**d**). The plots show the recovery of the fluorescence (normalized) after photobleaching. Scale bar: 10 μm. Data are presented as mean ± SD ($n = 9$ in **c** and $n = 10$ in **d**).

cell membrane recovered rapidly, pointing to their dynamic liquid-like property (Fig. 8d, e). These dynamic puncta, with their interaction with the cell membrane, are reminiscent of those natural ones formed via 2D LLPS along the cell membrane, such as semi-membrane-enclosed postsynaptic densities (PSDs) and endocytic vesicles[52–54].

**Light-induced liquid-to-solid phase transition in living cells**. To accomplish light-induced phase transition in living cells, the key is to deliver the photoreceptor, chlorophyll, into the cells. Spinach homogenates, which were dissolved in dimethyl sulfoxide (DMSO), were added into the cell cultures as the source of chlorophyll. Because of its lipophilicity, chlorophyll turned out to be able to cross the cell membranes and enter the cells efficiently, as evidenced by fluorescence imaging (Fig. 9b and Supplementary Fig. 17). The resulting cells were then transfected with the genes encoding RMR-mCherry and WSCP-EGFP. After culturing for 14–16 h, we observed the formation of protein condensates using fluorescence microscopy, during which these condensates were irradiated with 405-nm laser for ~5 min (15 mW). These condensates in the living cells exhibited good co-localization with chlorophyll, but not with WSCP-EGFP (Fig. 9b). The sizable EGFP fused with WSCP significantly altered the overall physical properties of this client molecule, which might account for the discrepancy in relation to the in vitro studies, where a neat WSCP construct was spontaneously recruited into the RMR condensates (Supplementary Fig. 4). The FRAP assay was further used to examine the dynamics of these irradiated condensates in the living cells. Contrary to those in the absence of WSCP and chlorophyll, of which the fluorescence quickly recovered to ~60% of their original about 1 min after photobleaching (Fig. 8c, d), the photobleached condensates in the presence of WSCP-EGFP and chlorophyll, both cytoplasmic and membrane-associated, barely recovered their fluorescence (<20%) after 1 min (Fig. 9c, d). The poor recovery of fluorescence strongly suggests the solid nature of these intracellular protein condensates. The realization of this chlorophyll-dependent, light-induced liquid-to-solid protein phase transition in living cells promises an alternative optical approach for interrogating intracellular signalling.

## Discussion

We have reported a red-light-induced liquid-to-solid phase transition system that can be employed to control the chemical reactivities of protein condensates and even to emulate their aging under physiological conditions. The key to this system is the decoration of the protein condensates with WSCP, a plant-derived chlorophyll-binding protein that generates singlet oxygen in response to red light. Incorporation of relatively hydrophilic WSCP stabilized the protein condensates and prevented their coalescence by reducing their surface tension in aqueous solutions. Moreover, singlet oxygen generated by WSCP under light irradiation triggered the phase transition of the protein condensates via oxidative crosslinking, a process that resembles the pathological protein aggregation in vivo often associated with aging and/or nerve degeneration. Unlike the natural process, where biological aging often occurs with a time scale of days or even years[10], our system can "age" precisely and rapidly; the use of optical tweezers enabled the spatiotemporal modulation of these protein condensates at the single-condensate level, within a timespan of a few minutes or even seconds, thus offering a sophisticated platform to study the interplay between biological aging and regulation.

As to the biofunctionalization of synthetic MLOs, we developed a strategy based on metal/His6-tag coordination for immobilizing protein cargos onto the protein condensates. This method circumvents the additional genetic manipulation needed for recombinant protein as previously reported[19,55]. While a good number of studies have focused on enzymatic assembly or condensation for improved efficiency, this study has been eyeing an alternative mechanism for controlling catalysis within an MLO; the light-induced liquid-to-solid phase transition restricts the

diffusion of substrates into the enzyme-laden protein condensate, thus shutting down the biochemical reactions within.

It is also noteworthy that the RMR protein exhibited a robust 2D phase separation behavior along the cell membrane in living cells. The Mfp-3 domain, a putative water-resistant adhesive motif, has proven to be essential for the observed 2D phase separation. While the previous studies have mostly been focused on the binding of Mfp-3 to nonbiologic substrates, such as mica, iron sheet, polystyrene, or polymethylmethacrylate[25,56,57], this study suggests that Mfp-3, even in the absence of post-translationally modified DOPA moieties, might well be adhesive to biologic substrates such as the lipid bilayers of cell membranes. Since 2D protein phase separation on cell membranes is not only crucial for the formation of the semi-membrane-bound assemblies such as PSDs, but also for the interplay between those membrane-closed and membraneless organelles, ranging from cargo trafficking to junction connection, liposome delivery, and endocytic vesicle formation, this RMR-based system, with its engineerability, offers a opportunity for studying the roles of 2D phase separation in various biological processes[52,58–60].

Although several optogenetic tools have been adopted for controlling intracellular LLPS in a blue-light-dependent manner, the ability to induce liquid-to-solid phase transition in living cells via a red-light-responsive photoreceptor has yet to be developed[61,62]. The facile delivery of chlorophyll, a ubiquitous photoreceptor in plants, into mammalian cells has been encouraging, which might open up a new dimension for optogenetics, given the wide availability of chlorophyll and its broad responsiveness to blue (400–500 nm) and red (650–700 nm) light. Besides, under light irradiation, chlorophyll alone proved sufficient to trigger the phase transition of RMR in the living cells (Supplementary Fig. 18), suggesting that this small-molecule natural product might well act as a tool for optical control in biological systems, independent of WSCP.

Last but not least, despite their prevalence in nature and established biochemical profiles, plant-derived WSCPs have yet to be explored for biological applications. This study represents a rare example of using WSCP to control a delicate biochemical system. While the WSCP used in this study senses and responds to both blue light (400–500 nm) (Supplementary Fig. 3 and 19) and red light (650–700 nm) (Fig. 3), further red-shifted chlorophyll derivatives (>700 nm) are not uncommon in nature, holding the promise to extend optical control to the far-red light region[33]. For example, the recently designated chlorophyll f absorbs maximally at 706 nm, and bacteriochlorophyll b at ~800 nm[63–65]. The diversity of these chlorophyll cofactors, alongside powerful protein engineering, can be tapped to develop more red- or far-red-light-dependent optogenetic tools.

## Methods

**Gene construction**. The *mfp-3* gene was purchased as a gBlocks gene fragment from Integrated DNA Technologies. The gene encoding RGG was amplified from the construct pET-RGG purchased from Addgene (Plasmid #124929). The plasmid pQE80l::*His6-RGG-mfp-3-RGG* was constructed by inserting the *RGG* and *Mfp-3* genes in pQE80l::*AAA*[66] using the restriction sites including NheI/SalI, SacI/SpeI, and XhoI/KpnI. *Escherichia coli* strain DH5α was used for cloning and plasmid amplification. The plasmid pCAGGS::*RGG-mfp-3-RGG-mCherry* was constructed by GENEWIZ. All plasmids used in this study are summed up in Supplementary Table 1.

**Expression and purification of RMR**. *E. coli* strain BL21(DE3) harboring pQE80l::*His6-RGG-mfp-3-RGG* was grown in Luria broth (LB) containing 100 mg/L ampicillin at 37 °C till the optical density at 600 nm (OD600) reached 0.6-0.8. Protein expression was induced by adding 400 μM isopropyl β-D-1-thiogalactopyranoside (IPTG) at 16 °C. After overnight, cell cultures were harvested by centrifugation and resuspended in the lysis buffer [500 mM NaCl, 20 mM Tris, 0.1 mM phenylmethylsulfonyl fluoride (PMSF), pH 7.5]. Resuspended cells were disrupted by sonication, followed by centrifugation to collect the cell pellets containing the inclusion bodies of the RMR protein. The cell pellets were then

washed with the lysis buffer supplemented with 1 M urea. The resulting pellets were then resuspended and incubated with the lysis buffer containing 8 M urea at 30 °C for 1 hour, followed by another centrifugation to separate the supernatants from the insoluble fractions. The solubilized RMR protein was purified using Ni-nitrilotriacetic acid (NTA) affinity chromatography. Briefly, the supernatants were loaded into a Ni-NTA column, washed with five column volumes of the wash buffer (500 mM NaCl, 20 mM Tris, 25 mM imidazole, 8 M urea, pH 7.5), and then eluted using two-column volumes of the elution buffer (500 mM NaCl, 20 mM Tris, 500 mM imidazole, 8 M urea, pH 7.5). Protein purity was assessed by sodium dodecyl sulphate-polyacrylamide gel electrophoresis (SDS-PAGE). After purification, proteins were dialyzed against the storage buffer (500 mM NaCl, 20 mM Tris, pH 7.5) at an elevated temperature, 52 °C, to avoid phase separation. After dialysis, aliquots of the protein were flash-frozen with liquid nitrogen and stored in −80 °C.

**Expression and purification of caspase-3**. The caspase-3 enzyme, procaspase3, in this study is an engineered version of caspase-3 with two mutations, D9A and D28A. The plasmid, pET21b::*procaspase-3* was obtained as a gift from Prof. A. Clay Clark at North Carolina State University.

*E. coli* strain BL21(DE3) harboring pET21b::*procaspase3* was grown in LB supplemented with 100 mg/L ampicillin at 37 °C till OD600 reached 0.6–0.8. Protein expression was induced by addition of 200 μM IPTG at 30 °C for 4 h. Cells were harvested by centrifugation and resuspended in the lysis buffer (300 mM NaCl, 20 mM Tris, 0.1 mM PMSF, 1 mM 2-Mercaptoethanol, pH 7.5). The cells were disrupted by sonication, followed by centrifugation to collect the supernatants. The protein was then purified by a Ni-NTA column, the wash buffer (300 mM NaCl, 20 mM Tris, 25 mM imidazole, 1 mM 2-mercaptoethanol, pH 7.5), and the elution buffer (300 mM NaCl, 20 mM Tris, 500 mM imidazole, 1 mM 2-mercaptoethanol, pH 7.5). The protein purity was assessed using SDS-PAGE. The purified caspase-3 was then dialyzed against the storage buffer (300 mM NaCl, 20 mM Tris, pH 7.5) at 4 °C overnight. After dialysis, glycerol [20% (v/v)] was added into the protein solution which was then flash-frozen in liquid nitrogen and stored at −80 °C.

**Cloning, expression, purification and reconstitution of WSCP**. The gene encoding WSCP was synthesized and cloned into pET22b and pEGFP-N1 by GENEWIZ. *E. coli* strain BL21(DE3) harboring pET22b::*WSCP* was grown in LB supplemented with 100 mg/L ampicillin at 37 °C till OD600 reached 0.6–1.2. Protein expression was induced by addition of 500 μM IPTG at 37 °C for 4 h. Cell pellets were harvested by centrifugation.

To incorporate chlorophyll into WSCP, spinach leaves were mixed with the bacterial cell pellets and the lysis buffer (300 mM NaCl, 20 mM Tris, 0.1 mM PMSF, pH 7.5) at a mass ratio of about 8:1:8. The mixture was then homogenized in a household blender, followed by sonication. The resulting homogenates were further centrifuged to separate the supernatants from the insoluble fractions. The reconstituted protein was purified from the supernatants using the standard Ni-NTA chromatography, involving the wash buffer (300 mM NaCl, 20 mM Tris, 25 mM imidazole, pH 7.5) and the elution buffer (300 mM NaCl, 20 mM Tris, 500 mM imidazole, pH 7.5). The protein might further be purified using size exclusion chromatography (SEC). The protein purity was assessed using SDS-PAGE.

The purified protein was dialyzed against MilliQ water. After dialysis, it was either used freshly or stored at −80 °C after flash-freezing in liquid nitrogen.

**Phase separation and turbidity tests of RMR**. Frozen protein aliquots were thawed in a 52 °C water bath to ensure the solubility of the protein. The phase separation was initiated by moving the protein samples out of the water bath to room temperature (23 °C). Turbidity tests were performed by monitoring their absorbance at 600 nm using a UV-Vis spectrophotometer (NanoDrop 2000c; Thermo Fisher). Three independent experiments were performed to ensure consistency.

**Phase transition of RMR/WSCP**. To the phase separated RMR solution was added LvWSCP (5 μM). The liquid-to-solid phase transition process was triggered by red LED light (630 nm). The light intensity was measured by a power meter (PM100D-THORLABS) with a standard photodiode power sensor (S120VC-THORLABS).

**Light-heat cycling of RMR/WSCP condensates**. In each light-heat cycle, protein solutions containing the RMR/WSCP condensates was first irradiated with red light at 23 °C for 5 min, subsequently incubated in 52 °C water bath in the dark for 5 min, and then moved back to 23 °C in the dark for 5 min prior to the next cycle. Samples were taken from the solutions at the end of each step within the cycle for imaging analyses. Protein condensates were counted manually from the photographs taken.

**Modulation of protein condensates by optical tweezers**. The experimental platform integrates a dual-beam optical tweezer system with an additional laser source in an inverted microscope. The dual-trap optical tweezers and bright-field imaging are the built-in features of the optical-tweezer microscopy system (m-trap, LUMICKS). The 1064-nm laser (80 mW) was split into two via a polarizing beam

splitter and focused inside the sample cell by a 60× water immersion objective with a 1.2 numerical aperture (N.A.) for dual-beam trapping. A 532-nm laser (MLL-III-532, CNI) was introduced into the same microscope as the additional laser source for triggering the liquid-to-solid phase transition. The triggering laser was aligned with the trapping laser via the stereo double-layer-pathway of the optical tweezer microscope with a 750-nm long-pass dichroic mirror to focus at the center of sample chamber. Adjusted by ND filters in the beam path to generate 2.5-mW input, the power density at the focus of the triggering laser was estimated to be $3.2 \times 10^5$ W/cm$^2$.

**Protein immobilization onto RMR condensates.** To the RMR solution at 52 °C was added NiCl$_2$ (50 μM) or ZnSO$_4$ (50 μM) and His6-GFP (2.5 μM) or His6-caspase-3 (2.5 μM). For the control experiment, EDTA (2 mM) was added to RMR solution with NiCl$_2$ and His6-GFP. The mixture was incubated at 52 °C for 1 h and then cooled down to 23 °C to initiate phase separation. The resulting condensates were imaged using confocal microscopy.

**Photo-controlled catalysis within RMR/WSCP condensates.** To the caspase-3-laden RMR condensates was added WSCP (5 μM). The mixtures were then exposed to red light or kept in the dark as designated. The resulting condensates were separated from the solvent via centrifugation (20,000 g; 10 min) and then resuspended in the reaction buffer (20 mM HEPES, 20 mM EDTA, 50 mM DTT, pH 7.5) supplemented with 10 mM fluorogenic Caspase-3 Substrate IV (Sigma-Aldrich). The reactions were incubated in the dark for 2 h and then analyzed using fluorometric assays.

The reaction solutions (100 μL) were transferred to a 96-well TC-treated plate (Sangon Biotech) and measured by a Varioskan LUX multimode microplate reader (Thermo Fisher) with the excitation and emission wavelengths at 400 and 505 nm, respectively. At least three independent experiments have been performed to ensure consistency.

**Cell culture and transfection.** HEK293 [American Type Culture Collection (ATCC)] and Hela (ATCC) cells were cultured in DMEM (Sangon Biotech) supplemented with 10% (v/v) FBS (Sangon Biotech) and 1% (v/v) penicillin-streptomycin solution (Sangon Biotech) in a 5% CO$_2$ atmosphere at 37 °C and passaged every 3 days. At 70–80% confluence, cells were detached with 1 mL of 0.25% trypsin solution (Sangon Biotech) followed by addition of 5 mL of DMEM to neutralize the trypsin. Around 20,000 cells were further cultured for 36 h before transfection. Transfection was accomplished using the Lipofectamine 3000 reagent (Thermo Fisher) mixed with 1 μg of the plasmid, pCAGGS::*RGG-mfp-3-RGG-mCherry* or pcDNA::*RGG-gfp-RGG*, per dish, according to the manufacturer's transfection protocol. Transfected cells were further incubated under the standard culturing condition for 12–16 h before imaging.

**Delivery of chlorophyll into living cells.** Spinach extracts were prepared following a reported methodology[67]. Leaves from 500 g of fresh spinach were washed, boiled for 1 min, and then dried with towel paper. Acetone (800 mL) was added to immerse the spinach leaves. The mixture was stirred at room temperature in the dark for 4 h. The resulting homogenate was then filtered, concentrated under reduced pressure, and lyophilized to generate a black hygroscopic solid. It was stored in −20 °C before use.

The spinach extract was dissolved in DMSO at a concentration of 10 mg/ml. HEK293 cells with ~70% confluency were washed with PBS, followed by the addition of the pre-warmed DMEM culture medium (37 °C) supplemented with the spinach extract (0.25 mg/ml). The cells were cultured under the standard culturing condition for 6 h. The cells were then washed with PBS twice to remove the medium supplemented with the spinach extract, followed by the addition of fresh DMEM for imaging or transfection.

**Light-induced phase transition in HEK293 cells.** HEK293 cells harbouring chlorophyll and transfected with the genes encoding RMR-mCherry and WSCP-EGFP were cultured for 14–16 h before imaging. The phase transition of the RMR condensates in HEK293 cells were triggered by irradiation with a 405-nm laser beam from the confocal microscope (Nikon C2) for 5 min, followed by the FRAP assay to detect the dynamics of the RMR condensates in the cytoplasm and in the outer membrane.

**Fluorescence confocal microcopy and fluorescence recovery after photobleaching.** Cells were imaged using a confocal microscope (Nikon C2) with a 63X oil-immersion objective and incubated with an incubator (Chamlide TC) during imaging. For chlorophyll signal, imaging was conducted by the 405 nm laser. In the FRAP assays, photobleaching was accomplished with a 561-nm laser. The fluorescence intensities arising from GFP and mCherry were quantified using the NIS-Elements software.

**Statistics and reproducibility.** All data were derived from at least three independent experiments unless indicated in the figure legends. Statistical analyses were performed using unpaired, two-tailed Student's t-test via GraphPad Prism 8. NIS-Elements AR software was used for micrographs collection, and NIS-Elements AR software and ImageJ 1.49 were used for image analysis.

**Reporting summary.** Further information on research design is available in the Nature Research Reporting Summary linked to this article.

## Data availability
All data needed to evaluate the conclusions in the paper are present in the paper and/or the Supplementary Materials. The crystal structure of WSCP shown in Fig. 1 is available from the Protein Data Bank, with the PDB ID, 2DRE. Figure Source data are provided with this paper.

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

## Acknowledgements

The funding supports from the Ministry of Science and Technology to F.S. (2020YFA0908100), from Natural Science Foundation of China Excellent Young Scientists Fund to F.S. (22122707), from Guangdong Natural Science Foundation to F.S. (2019A1515011691), from the Science, Technology, and Innovation Commission of Shenzhen Municipality to F.S. (Shenzhen-Hong Kong-Macau S&T Program (Category C) #SGDX2020110309460101, Key Research Program #JCYJ20200109141241950, and Basic Research Program #JCYJ20190813094601656), from the Research Grants Council of Hong Kong SAR Government to F.S. (General Research Fund #16103519 and #16103421; Theme-based Research Scheme #T13-602/21-N), from the Hong Kong Innovation and Technology Commission—Midstream Research Programme for Universities—Standalone Project (MRP/012/20), from the State Key Laboratory of Molecular Neuroscience, HKUST, China, and in part from the Innovation and Technology Commission (ITCPD/17-9) are acknowledged.

## Author contributions

F.S. and M.L. conceived of the design. M.L. and B.P. designed and performed experiments. J.H. and D.X. offered technical support on optical tweezers. Y.X. prepared reconstituted WSCP. F.S and M.L analyzed data and wrote the manuscript. All authors reviewed the final manuscript and provided inputs

## Competing interests

The authors declare no competing interests.
