## [Peer Review File · Nature Communications]

REVIEWER COMMENTS

Reviewer #1 (Remarks to the Author):

This paper by M. Li et al. presents a recombinant protein construct that is able to undergo phase separation to form, in a temperature-dependent fashion, condensates resembling membrane-less organelles. Water-soluble chlorophyll protein (WSCP) attached to the surface of such condensates produces, upon irradiation, singlet oxygen, which de-swells the protein condensates and turns them into solid-like particles. Caspase-3 attached to the condensates via Ni-His tag coordination loses its activity upon exposure to singlet oxygen produced by WSCP, presumably because the enzyme's substrate cannot penetrate the solidified particles any more. When the same recombinant protein construct is introduced into human cells it is seen to undergo phase separation there as well, mostly along the cell membrane.

These are very clever combinations of recombinant proteins that then exhibit highly interesting new functions. WSCP is introduced as a very useful component to control the system by (far) red light. All this will meet the interest of a broad readership in and beyond the field of protein phase separation. The various condensates and their solidification upon illumination are indeed reminiscent of the behaviour of some natural membrane-less organelles. Whether this is indeed a valid model of aging processes remains to be seen. But the authors are appropriately cautious in drawing such parallels to phenomena in vivo.

(1) WSCP used to decorate the phase-separated condensates is thought to be attached to the latter by hydrophobic interactions. WSCP is formed in a very straightforward way and no information is given about its purification prior to adding it to the condensates.

Therefore it cannot be ruled out that it actually is the chlorophyll pigment that is attached to or dissolved in the hydrophobic domains of the condensates and that then upon illumination may be the source of singlet oxygen. This may not be the most likely but still be a possible scenario and, therefore, should be ruled out. There are easy ways of doing this, for instance by native gel electrophoresis to detect intact WSCP and quantitate non-bound chlorophyll in the system. It would probably even suffice to measure the circular dichroism in the visible range where, in this system, only chlorophyll will absorb and indicate whether all of it is bound to WSCP.

(2) It has not become clear to me how the binding of protein cargo occurs to the condensates via Ni-His tag complexes. Presumably, the tyrosine side chains of the mussel foot protein are thought to bind to the Ni moiety? The cargo proteins do not bind if Ni ions are omitted. This, however, still leaves the possibility that binding occurs via some electrostatic effect rather than Ni complex formation. This should be ruled out by control experiments with other positively charged ions and/or EDTA.

(3) The loss of caspase activity upon illumination of its complex with condensates and WSCP is explained by the liquid-to-solid phase transition of the protein condensates, rendering them impermeable for peptide substrate of caspase (line 259). As an alternative interpretation of this observation, the singlet oxygen produced by irradiated WSCP may have inactivated the enzyme by chemically damaging it. If this possibility is difficult to exclude experimentally, unless there is a way to solubilize the solidified protein condensates, it should at least be discussed.

(4) There are some presumably unwanted remnants of word-processing actions in the text that affect its clarity, for instance line 711 "...., but not into allow free a solid particle after the phase transition."

Reviewer #2 (Remarks to the Author):

The present work "Controlling synthetic membraneless organelles by a red-light-dependent

singlet oxygen-generating protein” reports artificial protein condensates, of which liquidity can be transformed into solid states by red-light. The authors designed a phase separable protein with two known phase separable proteins, RGG and Mfp (termed RMR). Mfp could be cross-linked by singlet oxygen, which was generated by red-light illumination of a tetrameric chlorophyll-binding protein, WSCP. Combining RMR condensates and WSCP, the authors demonstrated liquid-to-solid transition of RMR condensates by red-light. The authors also showed that caspase reactions inside condensates can be inhibited by liquid-to-solid transition by preventing caspase substrate uptake into solidified condensate. This work presents a new strategy to alter protein condensate property (between liquid and solid) by applying light-dependent singlet oxygen generation.

A major concern, however, is that the value of the present work is not properly justified. Only use of developed liquid-to-solid transition was to examine caspase reaction, and this is somewhat expected from solid condensates. Moreover, while the authors mentioned this work reminiscent of natural liquid-to-solid transition of protein condensates, natural transition (e.g. fibril formation) is very different from chemical cross-linking in this study. And this method (red-light induced liquid-to-solid transition of protein condensates) was not applicable in cells. Only condensate formation of RMR in cells (but no liquid-to-solid transition) was demonstrated. Overall, presented data are yet preliminary, and more information on what we can obtain from this system must be experimentally demonstrated to be published in Nat. Commun.

Other comments

- 1. WSCP coating on RMR droplets can be confirmed by confocal microscopy.**
- 2. RMR cross-linking by WSCP-red light can be further confirmed by SDS-PAGE of RMR condensates.**
- 3. Recruitment by 6His and metal ions here was relatively weak. More strong specific interactions can be introduced to RMR and used for client recruitment.**
- 4. An important advantage of light-mediated methods is spatial modulation. It would be great if only part of condensates can be solidified by focused light illumination.**
- 5. Again the ‘RMR condensate in living cells’ section is not relevant to ‘Controlling synthetic membraneless organelles’. Light-induced liquid-to-solid transition must be shown in cells.**

Reviewer #3 (Remarks to the Author):

This report describes the use of a plant-derived protein in combination with a liquid-liquid phase separating protein system to achieve optical control over the solid/liquid nature of the condensates. While the condensates alone undergo LCST-like phase separation, exposure to red light in the presence of the WSCP protein resulted in crosslinking, and therefore solidification, of the condensates due to the production of singlet oxygen. The authors further demonstrate that this optical trigger and the resulting crosslinking can be used to modulate the activity of enzymes entrained within the condensates, ostensibly by limiting diffusion of the substrate. While the general idea presented here is very interesting, I have questions about the mechanism whereby enzymatic activity is lost. Addressing these points would help to strengthen the potential impact of the article. I have also included several other more minor suggestions.

Specific Comments:

- 1. The authors propose that the decrease in enzymatic activity observed upon solidification of the condensates is due to an inability of substrate to diffuse. Could they please speculate on the size limits of this diffusional barrier? Furthermore, how do we know that the loss in activity is related only to the accessibility of the substrate, and not to a damaging of either the enzymes or the substrate itself in the presence of singlet oxygen? Perhaps an analysis of the enzyme kinetics would provide some insight into this question, along with FRAP or other diffusional studies.**
- 2. Would it be possible to create a fluorescently tagged WSCP to enable direct imaging in support of the idea that the protein is localized on the surface of the condensates?**
- 3. Would it make sense to include a depiction of an enzyme within the droplet schematics shown in Figure 1?**

HONG KONG UNIVERSITY OF SCIENCE AND TECHNOLOGY

DEPARTMENT OF CHEMICAL AND BIOLOGICAL ENGINEERING

Fei SUN

Phone: +852 3469 2442

Associate Professor of Chemical and Biological Engineering

FAX: +852 2358 0054

kefsun@ust.hk

March 22, 2022

Dear Editor:

Thank you for handling our manuscript. We also thank the reviewers for their thoughtful comments and suggestions. We have considered the comments carefully, have performed several additional experiments, and have addressed the major issues raised by the reviewers, including elucidation of the various molecular mechanisms (e.g., the source of singlet oxygen, cargo recruitment, and inhibition of enzymatic activities), optical modulation of the protein condensates with high spatial precision, and light-induced liquid-to-solid phase transition in living cells.

Our point-by-point response and the revision of the manuscript have been elaborated as follows:

Response to Comments of Reviewer 1:

This paper by M. Li et al. presents a recombinant protein construct that is able to undergo phase separation to form, in a temperature-dependent fashion, condensates resembling membrane-less organelles. Water-soluble chlorophyll protein (WSCP) attached to the surface of such condensates produces, upon irradiation, singlet oxygen, which de-swells the protein condensates and turns them into solid-like particles. Caspase-3 attached to the condensates via Ni-His tag coordination loses its activity upon exposure to singlet oxygen produced by WSCP, presumably because the enzyme's substrate cannot penetrate the solidified particles anymore. When the same recombinant protein construct is introduced into human cells it is seen to undergo phase separation there as well, mostly along the cell membrane.

These are very clever combinations of recombinant proteins that then exhibit highly interesting new functions. WSCP is introduced as a very useful component to control the system by (far) red light. All this will meet the interest of a broad readership in and beyond the field of protein phase separation. The various condensates and their solidification upon illumination are indeed reminiscent of the behaviour of some natural membrane-less organelles. Whether this is indeed a valid model of aging processes remains to be seen. But the authors are appropriately cautious in drawing such parallels to phenomena *in vivo*.

Comment 1.1 WSCP used to decorate the phase-separated condensates is thought to be attached to the latter by hydrophobic interactions. WSCP is formed in a very straightforward way and no information is given about its purification prior to adding it to the condensates. Therefore, it cannot be ruled out that it actually is the chlorophyll pigment that is attached to or dissolved in the hydrophobic domains of the condensates and that then upon illumination may be the source of singlet oxygen. This may not be the most likely but still be a possible scenario and, therefore, should be ruled out. There are easy ways of doing this, for instance by native gel electrophoresis to detect intact WSCP and quantitate non-bound chlorophyll in the system. It would probably even suffice to measure the circular dichroism in the visible range where, in this system, only chlorophyll will absorb and indicate whether all of it is bound to WSCP.

Response: The reconstituted WSCP used for the decoration of the condensates was purified from *Escherichia coli* lysates first by Ni-NTA chromatography and then by size-exclusion chromatography (SEC), to ensure its purity. The SEC data showed that the reconstituted WSCP was predominantly tetrameric, highly indicative of the formation of the chlorophyll/WSCP complex (Supplementary Fig. 4 in revised SI). This complex has previously been shown to be remarkably stable toward dissociation and protein denaturation even at 100 °C and extreme pH values (pH 0-14) [Palm et al (2017) *Biochemistry*]. Therefore, we can rule out the possibility that free chlorophyll can dissociate from the complex under the ambient conditions (i.e., at room temperature, pH 7-8) used in this study, not to mention the generation of singlet oxygen by free chlorophyll.

The manuscript (Page 5-6) has been revised to include the discussion above.

Supplementary Fig 8. SEC analyses of the reconstituted chlorophyll-binding WSCP (tetrameric) and apoWSCP (monomeric).

Reference:

Palm, D.M., et al., Water-Soluble Chlorophyll Protein (WSCP) Stably Binds Two or Four Chlorophylls. *Biochemistry*, 2017. 56(12): p. 1726-1736.

Comment 1.2 It has not become clear to me how the binding of protein cargo occurs to the condensates via Ni-His tag complexes. Presumably, the tyrosine side chains of the mussel foot protein are thought to bind to the Ni moiety? The cargo proteins do not bind if Ni ions are omitted. This, however, still leaves the possibility that binding occurs via some electrostatic effect rather than Ni complex formation. This should be ruled out by control experiments with other positively charged ions and/or EDTA.

Response: In our system, His6-tag, a putative transition metal-binding ligand that is present in both RMR and the protein cargos, is mainly responsible for the Ni²⁺-dependent binding of protein cargos into the condensates, while other ligands such as tyrosine likely play a minor role, if there is, in the binding of protein cargos (Supplementary Fig. 1). Several previous studies have demonstrated the use of metal/His6-tag coordination as a versatile strategy to modulate protein materials [Jiang et al (2020) *Science advances* , Kou et al (2019) *ACS Macro Letters*] or to generate protein condensates or decorate them with functional His6-tagged protein motifs [Hong et al (2020) *Nature communication*].

To confirm that the binding of protein cargos occurs mainly via metal/ligand coordination, rather than some other forces such as electrostatic effect, we examined the influence of ethylenediaminetetraacetic acid (EDTA) and transition metal ions other than Ni²⁺ on the interactions between His6-tagged GFP and the protein condensates. It turned out that the presence of excess of EDTA (2 mM) led to the exclusion of GFP from the Ni²⁺-laden protein condensates, comparable to that in the absence of the divalent metal ions (revised Fig. 6b-c). Other transition metal ions such

as Zn^{2+} also facilitated the immobilization of His6-tagged GFP onto the RMR condensates (Supplementary Fig. 13). Together, these results suggested that, among many possible interactions, the metal/ligand coordination is the chief one that drives the recruitment of His6-tagged protein cargos into the RMR condensates.

Fig. 6. Decoration of RMR condensates with His6-tagged GFP via metal coordination. **a** Schematic illustration of Ni^{2+} -induced recruitment of His6-tagged GFP into RMR condensates. **b** Fluorescence micrographs showing GFP-depleted and GFP-laden RMR condensates in the absence and presence of Ni^{2+} (50 μM), as well as those in the presence of Ni^{2+} (50 μM) and excess of EDTA (2 mM). Scale bars: 10 μm . **c** Relative distribution of GFP inside and outside RMR condensates. Enrichment efficiency was calculated using the equation, $Enrichment = \frac{\frac{GFP(in,+Ni)}{GFP(out,+Ni)}}{\frac{GFP(in,-Ni)}{GFP(out,-Ni)}}$.

Data are presented as mean \pm SD ($n = 10$); p-value: ****<0.0001. **d** Representative GFP-laden condensate and its normalized fluorescence intensity profile. Scale bar: 1 μm .

Supplementary Fig. 13. Fluorescence micrographs showing GFP-laden RMR condensates in the presence of Zn^{2+} (50 μM). Scale bars: 10 μm .

References:

Jiang, B., et al., Injectable, photoresponsive hydrogels for delivering neuroprotective proteins enabled by metal-directed protein assembly. *Science advances* 6.41 (2020): eabc4824.

Kou, S., et al., Cobalt-cross-linked, redox-responsive spy network protein hydrogels. *ACS Macro Letters* 8.7 (2019): 773-778.

Hong, K., et al., Behaviour control of membrane-less protein liquid condensates with metal ion-induced phase separation. *Nature communications* 11.1 (2020): 1-12.

Comment 1.3 The loss of caspase activity upon illumination of its complex with condensates and WSCP is explained by the liquid-to-solid phase transition of the protein condensates, rendering them impermeable for peptide substrate of caspase (line 259). As an alternative interpretation of this observation, the singlet oxygen produced by irradiated WSCP may have inactivated the enzyme by chemically damaging it. If this possibility is difficult to exclude experimentally, unless there is a way to solubilize the solidified protein condensates, it should at least be discussed.

Response: To rule out the possibility that singlet oxygen may directly compromise the caspase-3 activity by chemically damaging the enzyme, we examined the activities of free caspase-3 in the presence of WSCP under red light illumination (5 mW/cm²). After 30-min irradiation, the enzymatic activities were barely affected, suggesting the negligible influence of singlet oxygen on free caspase-3 itself (Supplementary Fig. 15). This result pointed to the liquid-to-solid phase transition induced by photo-oxidation as a more plausible mechanism that accounts for the loss of the caspase activity within the protein condensates.

The manuscript (page 9) has been revised to include this discussion and the new Supplementary Fig. 15.

Supplementary Fig. 15. Negligible influence of singlet oxygen generated by WSCP on free caspase-3 after 30-min red-light irradiation. Hydrolysis of the fluorogenic substrate for caspase-3, Z-DEVD-AFC, which was accompanied with an increased emission at 500 nm (ex. 405 nm), served as an indicator of the caspase-3 activity. **a** Normalized fluorescence intensities showing the influence of red-light irradiation on free caspase-3 in the solutions in the presence and absence of WSCP. **b** Normalized fluorescence intensities showing the influence of red-light irradiation on bound caspase-3 within RMR condensates in the presence and absence of WSCP.

Light intensity: 5 mW/cm². Data are presented as mean ± SD ($n = 3$); p-value: *<0.05; ****<0.0001.

(4) There are some presumably unwanted remnants of word-processing actions in the text that affect its clarity, for instance line 711 „...“, but not into allow free a solid particle after the phase transition.

Response: These texts have been corrected. Thanks.

Response to Comments of Reviewer 2:

The present work “Controlling synthetic membraneless organelles by a red-light-dependent singlet oxygen-generating protein” reports artificial protein condensates, of which liquidity can be transformed into solid states by red-light. The authors designed a phase separable protein with two known phase separable proteins, RGG and Mfp (termed RMR). Mfp could be cross-linked by singlet oxygen, which was generated by red-light illumination of a tetrameric chlorophyll-binding protein, WSCP. Combining RMR condensates and WSCP, the authors demonstrated liquid-to-solid transition of RMR condensates by red-light. The authors also showed that caspase reactions inside condensates can be inhibited by liquid-to-solid transition by preventing caspase substrate uptake into solidified condensate. This work presents a new strategy to alter protein condensate property (between liquid and solid) by applying light-dependent singlet oxygen generation.

A major concern, however, is that the value of the present work is not properly justified. Only use of developed liquid-to-solid transition was to examine caspase reaction, and this is somewhat expected from solid condensates. Moreover, while the authors mentioned this work reminiscent of natural liquid-to-solid transition of protein condensates, natural transition (e.g. fibril formation) is very different from chemical cross-linking in this study. And this method (red-light induced liquid-to-solid transition of protein condensates) was not applicable in cells. Only condensate formation of RMR in cells (but no liquid-to-solid transition) was demonstrated. Overall, presented data are yet preliminary, and more information on what we can obtain from this system must be experimental demonstrated to be published in Nat. Commun.

Response: We thank the reviewer’s constructive inputs. The natural liquid-to-solid transition of protein condensates can be very complex, some of which, such as amyloid fibril formation, is associated not only with the changes in protein secondary structures and their material properties, but also with chemical crosslinking such as disulfide bond [Honda, R (2018) *Biophys J*], dityrosine crosslink [Mukherjee et al (2017) *Anal Chem*], etc. In this study, we are trying to develop a de novo designed system not only to recapitulate several, if not all, important aspects (i.e., materials properties, covalent crosslinking) that are featured in those natural protein phase transition processes, but also to achieve controllability that remains elusive for the natural systems. In the revised manuscript, we have managed to demonstrate the feasibility of using this light-inducible system to control liquid-to-solid phase transition in living cells (Fig. 9), pointing to a new optogenetic approach for controlling intracellular protein phase transition. Moreover, the ability to control the intracellular protein phase transition with red light is also advantageous, especially when in vivo applications and deep tissue penetration are concerned, when compared to the existing methods that often rely on short-wavelength light such as blue light.

References:

Honda, R., Role of the Disulfide Bond in Prion Protein Amyloid Formation: A Thermodynamic and Kinetic Analysis. *Biophys J*, 2018. 114(4): p. 885-892.

Mukherjee, S., et al., Characterization and Identification of Dityrosine Cross-Linked Peptides Using Tandem Mass Spectrometry. *Anal Chem*, 2017. 89(11): p. 6136-6145.

Other comments:

Comment 2.1 WSCP coating on RMR droplets can be confirmed by confocal microscopy.

Response: The cofactor of WSCP, chlorophyll, is intrinsically fluorescent [Maxwell et al (2000) *Journal of experimental botany*], which allowed us to assess the distribution of WSCP within the RMR condensates using confocal microscopy (Supplementary Fig. 7). The confocal imaging results revealed an obvious enrichment of WSCP by the RMR condensates, likely due to the hydrophobic effect. It appears to us that WSCP was present not only on the surfaces but also inside of the condensates. The quantitative analysis on the exact distribution of WSCP within a condensate remains challenging, which is largely constrained by the small size of the condensate (< 5 μm in diameter) and the limited resolution of the confocal microscope. Nevertheless, it is still safe to say that the WSCPs on the surfaces of these condensates are crucial for the reduced surface tension and the stabilization of these condensates.

Supplementary Fig 4. Confocal images of RMR/WSCP condensates. a Enrichment of WSCP by RMR condensates. Ex.: 405 nm; Em.: 640 nm. TD, transmitted detector image. Scale bar: 10 μm . **b** 3D rendering and representative z-slice images of RMR/WSCP condensates. Scale bar: 1 μm .

Reference:

Maxwell, K. and G.N. Johnson, Chlorophyll fluorescence—a practical guide. *Journal of experimental botany*, 2000. 51(345): p. 659-668.

Comment 2.2 RMR cross-linking by WSCP-red light can be further confirmed by SDS-PAGE of RMR condensates.

Response: The newly performed SDS-PAGE analysis revealed some additional bands in the presence of WSCP after red-light irradiation, corresponding to the higher-MW crosslinked products. These products were absent under the dark conditions. The amount of monomeric RMR decreased as the duration of light illumination increased. This result confirmed the RMR crosslinking in the presence of WSCP under red light.

The manuscript (Page 6) has been revised to include this result as Supplementary Fig. 8.

Supplementary Fig 6. SDS-PAGE analysis of RMR/WSCP condensates under light and dark conditions. Some crosslinked products are highlighted in a dash-line box.

Comment 2.3 Recruitment by 6His and metal ions here was relatively weak. More strong specific interactions can be introduced to RMR and used for client recruitment.

Response: Several previous studies have demonstrated that strong specific interactions, such as SpyTag/SpyCatcher chemistry, a peptide/protein pair that can spontaneously form a Lys-Asp isopeptide bond, can be used to decorate protein materials such as hydrogels and synthetic membraneless organelles [Kirst et al (2022) *Proceedings of the National Academy of Sciences*, Liu et al (2019) *ACS nano*, Sun et al (2014) *Proceedings of the National Academy of Sciences*]. Despite their strength in specificity and efficiency, these approaches require special design for the protein constructs to be introduced to the condensates.

The metal/His6 coordination circumvents the trouble of genetically fusing the proteins with specific peptide/protein motifs, as the His6-tag used has already been prevalent among recombinant proteins for purification purposes. The metal/His coordination, though relatively weak for free proteins in aqueous solutions, has proven to be very efficient in recruiting client molecules into the protein condensates (enrichment efficiency of ~8.5 fold) in this study (Fig. 6b-c), also in agreement with a previous study [Hong et al (2020) *Nature communication*], which can partly be explained by

the unusually high local concentration of His6-tag within the condensed phase. Therefore, the method presented in this study is efficient for the recruitment of various recombinant proteins into the protein condensates, and more generalizable than those based on specific protein/protein interactions.

References:

Kirst, H., et al., Toward a glycol radical enzyme containing synthetic bacterial microcompartment to produce pyruvate from formate and acetate. *Proceedings of the National Academy of Sciences*, 2022. 119(8): p. e2116871119.

Liu, Z., et al., Self-assembled multienzyme nanostructures on synthetic protein scaffolds. *ACS nano*, 2019. 13(10): p. 11343-11352.

Sun, F., et al., Synthesis of bioactive protein hydrogels by genetically encoded SpyTag-SpyCatcher chemistry. *Proceedings of the National Academy of Sciences*, 2014. 111(31): p. 11269-11274.

Hong, K., et al., Behavior control of membrane-less protein liquid condensates with metal ion-induced phase separation. *Nature communications* 11.1 (2020): 1-12.

Comment 2.4 An important advantage of light-mediated methods is spatial modulation. It would be great if only part of condensates can be solidified by focused light illumination.

Response: We have achieved spatial modulation of the RMR/WSCP condensates using optical tweezers. We used a modified dual-beam optical tweezer system to trap two protein condensates (Trap 1 and Trap 2), respectively, followed by irradiating the designated site of the chosen condensate (Trap 1; *left*) with an additional triggering laser beam to trigger the liquid-to-solid phase transition (Fig. 5 and Supplementary Fig. 12).

The trapped condensate (Trap 1) started to shrink—a sign of solidification—first at its irradiated site within 7 s and then across the entire condensate, resulting in an irregular solid particle within 15 s, while the control (Trap 2) without laser irradiation remained intact throughout the experiment. This result demonstrated the possibility of optically controlling these protein condensates at the single-condensate level, with high spatial precision.

The results are now present in Fig. 5 and Supplementary Fig. 12 in the revised manuscript.

Fig. 5. Modulation of RMR/WSCP condensates with high spatial precision by optical tweezers. **a** Schematic illustration of the use of a modified dual-beam optical tweezer to modulate the protein condensates at the single-condensate level. Trapping laser: 1064 nm; 80 mW. Triggering laser: 532 nm; 2.5 mW. **b** Schematic illustration of the trapping of two protein condensates (Trap 1 and Trap 2) by optical tweezers, followed by selective irradiation of the chosen condensate (Trap 1; left) via the triggering laser beam. A live camera image of two trapped condensates is shown. Scale bar: 5 μm . **c** Liquid-to-solid phase transition of the chosen RMR/WSCP condensate (*left*) under the irradiation of the triggering laser beam within 15 s. The concentration of WSCP is 5 μM . Scale bar: 5 μm . **d** The RMR condensate in the absence of WSCP is inert to prolonged irradiation. Scale bar: 1 μm .

Supplementary Fig. 12. Liquid-to-solid phase transition of a trapped RMR/WSCP condensate (*left*) under the irradiation of a triggering laser beam. The concentration of WSCP is 1 μM . Scale bar: 5 μm .

Comment 2.5 Again the ‘RMR condensate in living cells’ section is not relevant to ‘Controlling synthetic membraneless organelles’. Light-induced liquid-to-solid transition must be shown in cells.

Response: In the revised manuscript, we have managed to demonstrate the light-induced liquid-to-solid phase transition in HEK293 cells (Fig. 9). The delivery of the photoreceptor, chlorophyll, into cells turned out to be straightforward. Spinach extracts, which served as a source of chlorophyll, were dissolved in DMSO and then added into the cell culture medium. Chlorophyll, because of its lipophilicity, was able

to cross the cell membrane and enter the cells efficiently, as evidenced by fluorescence imaging (Fig. 9b and Supplementary Fig. 17). The resulting cells were then transfected with the genes encoding RMR-mCherry and WSCP-EGFP. After the incubation for 14-16 h, we observed the formation of protein condensates using fluorescence microscopy, during which these condensates were irradiated with 405-nm laser for ~ 5 min (15 mW). FRAP was further used to examine the dynamics of these condensates. Contrary to those in the cells free of WSCP and chlorophyll, of which the fluorescence, after photobleaching, quickly recovered to ~50-60% of their original within ~1 min (Fig. 8c-d), the photobleached condensates, both cytoplasmic and membrane-bound, barely recovered their fluorescence even after 1 min (Fig. 9c, d). The poor recovery of fluorescence is highly indicative of the formation of solid-like protein condensates in the living cells under light irradiation. Together, these results demonstrated the feasibility of optically inducing liquid-to-solid phase transition of the protein condensates in living cells, which points to an alternative chlorophyll-dependent, optogenetic approach for studying intracellular protein phase transition.

The manuscript has been revised to include these results.

Fig. 9. Light-induced liquid-to-solid phase transition in HEK293 cells. **a** Schematic illustration of light-induced phase transition in cells. **b** Fluorescent images of the cells harbouring chlorophyll and expressing RMR-mCherry and EGFP-WSCP. Scale bar: 10 μm. **c** and **d** FRAP assays of RMR condensates in the cytoplasm (**c**) and in the outer membrane of HEK293 cells (**d**). The plots show the recovery of the fluorescence (normalized) after photobleaching. Scale bar: 10 μm. Data are presented as mean ± SD ($n = 9$ in **c** and $n = 10$ in **d**).

Response to Comments of Reviewer 3:

This report describes the use of a plant-derived protein in combination with a liquid-liquid phase separating protein system to achieve optical control over the solid/liquid nature of the condensates. While the condensates alone undergo LCST-like phase separation, exposure to red light in the presence of the WSCP protein resulted in crosslinking, and therefore solidification, of the condensates due to the production of singlet oxygen. The authors further demonstrate that this optical trigger and the resulting crosslinking can be used to modulate the activity of enzymes entrained within the condensates, ostensibly by limiting diffusion of the substrate. While the general idea presented here is very interesting, I have questions about the mechanism whereby enzymatic activity is lost. Addressing these points would help to strengthen the potential impact of the article. I have also included several other more minor suggestions.

Specific Comments:

Comment 3.1 The authors propose that the decrease in enzymatic activity observed upon solidification of the condensates is due to an inability of substrate to diffuse. Could they please speculate on the size limits of this diffusional barrier? Furthermore, how do we know that the loss in activity is related only to the accessibility of the substrate, and not to a damaging of either the enzymes or the substrate itself in the presence of singlet oxygen? Perhaps an analysis of the enzyme kinetics would provide some insight into this question, along with FRAP or other diffusional studies.

Response: According to the FRAP assays (Fig. 9), the solidification of the RMR condensates induced by photo-oxidation has drastically impeded the fluorescence recovery, reflecting the inability of external molecules to diffuse in. In addition, as elaborated in our Response to Comment 1.3, we also examined the activities of free caspase-3 in the presence of WSCP under red light illumination (5 mW/cm^2). After 30-min irradiation, the enzymatic activities were barely affected, suggesting the negligible influence of singlet oxygen on the free enzyme (Supplementary Fig. 15). Together, these results pointed to the liquid-to-solid phase transition induced by photo-oxidation as a more plausible mechanism for the loss of the caspase activity within the protein condensates.

Supplementary Fig. 15. Negligible influence of singlet oxygen generated by WSCP on free caspase-3 under light irradiation. Hydrolysis of the fluorogenic substrate for caspase-3, Z-DEVD-AFC, which was accompanied with an increased

emission at 500 nm (ex. 405 nm), served as an indicator of the caspase-3 activity. **a** Normalized fluorescence intensity showing the activity of free caspase-3 in the solution. **b** Normalized fluorescence intensity showing the activity of caspase-3 recruited in the RMR condensates. Light intensity: 5 mW/cm². Red-light illumination duration: 30 min. Data are presented as mean ± SD (*n* = 3); p-value: *<0.05; ****<0.0001.

Comment 3.2 Would it be possible to create a fluorescently tagged WSCP to enable direct imaging in support of the idea that the protein is localized on the surface of the condensates?

Response: The cofactor of WSCP, chlorophyll, is intrinsically fluorescent [Maxwell et al (2000) *Journal of experimental botany*], which allowed us to visualize WSCP within the RMR condensates using confocal microscopy (Supplementary Fig. 7). The confocal imaging results revealed an obvious enrichment of WSCP by the RMR condensates, likely due to hydrophobic effect. It appears to us that WSCP was present not only on the surfaces but also inside of the condensates. The quantitative analysis on the exact distribution of WSCP within a condensate remains quite challenging, which is largely constrained by the small size of the condensate (< 5 μm in diameter) and the limited resolution of the confocal microscope. Nevertheless, it is safe to say that the WSCP on the surface of these condensates is mainly responsible for the reduced surface tension and the stabilization of these condensates.

Supplementary Fig 4. Confocal images of RMR/WSCP condensates. a Enrichment of WSCP by RMR condensates. Ex.: 405 nm; Em.: 640 nm. TD, transmitted detector image. Scale bar: 10 μm. **b** 3D rendering and representative z-slice images RMR/WSCP condensates. Scale bar: 1 μm.

Reference:

Maxwell, K. and G.N. Johnson, Chlorophyll fluorescence—a practical guide. *Journal of experimental botany*, 2000. 51(345): p. 659-668.

Comment 3.3 Would it make sense to include a depiction of an enzyme within the droplet schematics shown in Figure 1?

Response: The figure has been revised to include the depiction of enzymes. Thanks.

Fig. 1. Schematic illustration of the red-light-controlled protein condensate enabled by water-soluble chlorophyll protein (WSCP). The protein condensate formed via LLPS of the recombinant protein, RGG-Mfp-3-RGG (RMR), can be decorated with WSCP, a tetrameric protein that generates singlet oxygen (1O_2) under red light illumination. Upon oxidative crosslinking by 1O_2 , the protein condensate undergoes a liquid-to-solid phase transition, which restricts the diffusion of substrates into the enzyme-laden protein condensate and turns off the catalysis within.

REVIEWERS' COMMENTS

Reviewer #1 (Remarks to the Author):

All concerns have been carefully addressed. I have no further comments.

Reviewer #2 (Remarks to the Author):

The authors properly addressed raised comments. In particular, added in-cell demonstration of the system (Figure 9) and optical tweezer experiments for spatial regulation (Figure 5) clearly strengthen the article. I believe that it is now suitable for publication.

One minor remaining point, however, is that WSCP looks to be inside RMR droplets in Figure S4. I don't see any problem on stating 'WSCP is both inside droplets and on surfaces', instead of stating 'WSCP coating on droplets', which is still not confirmed properly.

Reviewer #3 (Remarks to the Author):

I very much appreciate the improvements and additions to this new version of the manuscript. In particular, the control experiment demonstrating that caspase-3 activity is unaffected by the presence of singlet oxygen helps to confirm the presented interpretation of the results. Furthermore, the demonstration of in vivo solidification enhanced the novelty and utility of this report. I very much look forward to the publication of this manuscript.

HONG KONG UNIVERSITY OF SCIENCE AND TECHNOLOGY

DEPARTMENT OF CHEMICAL AND BIOLOGICAL ENGINEERING

Fei SUN

Phone: +852 3469 2442

Associate Professor of Chemical and Biological Engineering FAX: +852 2358 0054

kefsun@ust.hk

April 28, 2022

Dear Editor:

Thank you so much for your prompt handling of our revised manuscript (NCOMMS-22-00957A). We also thank all three reviewers for their positive comments on our study.

After considering the minor point raised by Reviewer 2, we have revised the manuscript correspondingly to clarify the issue. A detailed response has been enclosed below. The entire manuscript has also been reformatted according to the Author Guidelines and Editorial Requests. I hope that you will find this revised manuscript ready for its publication in Nature Communications.

Sincerely,

Fei Sun

Reviewer #2 (Remarks to the Author):

The authors properly addressed raised comments. In particular, added in-cell demonstration of the system (Figure 9) and optical tweezer experiments for spatial regulation (Figure 5) clearly strengthen the article. I believe that it is now suitable for publication.

One minor remaining point, however, is that WSCP looks to be inside RMR droplets in Figure S4. I don't see any problem on stating 'WSCP is both inside droplets and on surfaces', instead of stating 'WSCP coating on droplets', which is still not confirmed properly.

Response: We agree with the Reviewer on this issue. We have revised the caption of Figure 3c and the maintext to clearly state the presence of WSCP both inside droplets and on surfaces.